# A saturation-mutagenesis analysis of the interplay between stability and activation in Ras

**Frank Hidalgo**[1,2,3†], **Laura M Nocka**[1,2,3†], **Neel H Shah**[1,2,4,5], **Kent Gorday**[1,2,6], **Naomi R Latorraca**[1,4], **Pradeep Bandaru**[1,2,4], **Sage Templeton**[1,2], **David Lee**[1,2], **Deepti Karandur**[1,2,4], **Jeffrey G Pelton**[1], **Susan Marqusee**[1,3,4], **David Wemmer**[1,3], **John Kuriyan**[1,2,3,4]*

[1]California Institute for Quantitative Biosciences (QB3), University of California, Berkeley, Berkeley, United States; [2]Howard Hughes Medical Institute, University of California, Berkeley, Berkeley, United States; [3]Department of Chemistry, University of California, Berkeley, Berkeley, United States; [4]Department of Molecular and Cell Biology, University of California, Berkeley, Berkeley, United States; [5]Department of Chemistry, Columbia University, New York, United States; [6]Biophysics Graduate Group, University of California, Berkeley, Berkeley, United States

**\*For correspondence:**
kuriyan@berkeley.edu

†These authors contributed equally to this work

**Abstract** Cancer mutations in Ras occur predominantly at three hotspots: Gly 12, Gly 13, and Gln 61. Previously, we reported that deep mutagenesis of H-Ras using a bacterial assay identified many other activating mutations (Bandaru et al., 2017). We now show that the results of saturation mutagenesis of H-Ras in mammalian Ba/F3 cells correlate well with the results of bacterial experiments in which H-Ras or K-Ras are co-expressed with a GTPase-activating protein (GAP). The prominent cancer hotspots are not dominant in the Ba/F3 data. We used the bacterial system to mutagenize Ras constructs of different stabilities and discovered a feature that distinguishes the cancer hotspots. While mutations at the cancer hotspots activate Ras regardless of construct stability, mutations at lower-frequency sites (e.g. at Val 14 or Asp 119) can be activating or deleterious, depending on the stability of the Ras construct. We characterized the dynamics of three non-hotspot activating Ras mutants by using NMR to monitor hydrogen-deuterium exchange (HDX). These mutations result in global increases in HDX rates, consistent with destabilization of Ras. An explanation for these observations is that mutations that destabilize Ras increase nucleotide dissociation rates, enabling activation by spontaneous nucleotide exchange. A further stability decrease can lead to insufficient levels of folded Ras – and subsequent loss of function. In contrast, the cancer hotspot mutations are mechanism-based activators of Ras that interfere directly with the action of GAPs. Our results demonstrate the importance of GAP surveillance and protein stability in determining the sensitivity of Ras to mutational activation.

## Editor's evaluation

This is a well executed study that provides significant new insights and connections between protein structure, stability and function. Numerous sites of mutation are identified that activate the small GTPase Ras beyond the few cancer "hot spots" that predominate cancer genomics data. While cancer causing mutations selectively alter regulatory interactions with GTPase-activating proteins, careful biophysical analysis presented here leads to the conclusion that Ras is also activated by mutations that decrease stability (increase dynamics) short of unfolding. Thus, protein sensitivity to activating mutations can depend on the stability threshold in addition to regulatory interactions with binding partners.

## Introduction

The small GTPase Ras (*Figure 1A*) cycles between active GTP-bound and inactive GDP-bound states (*Figure 1B*; *Wittinghofer and Vetter, 2011*). There are four principal isoforms of human Ras (H-Ras, N-Ras, and two splice variants of K-Ras; referred to collectively as 'Ras'). GTP-bound Ras binds to the Ras-binding domains (RBDs) of effector proteins, such as Raf kinases and PI-3 kinase (PI3K), triggering signaling cascades that result in cell proliferation (*Ehrhardt et al., 2002*; *Pylayeva-Gupta et al., 2011*; *Schubbert et al., 2007*). Ras proteins have weak intrinsic GTPase activity (*Ehrhardt et al., 2002*). In the cell, Ras activity is controlled by two kinds of regulators: GTPase-activating proteins (GAPs) and guanine nucleotide-exchange factors (GEFs). GAPs stimulate the hydrolysis of GTP, thereby converting Ras to the inactive GDP-bound state (*Ahmadian et al., 1997*). Spontaneous exchange of GDP for GTP is slow, and nucleotide exchange and re-activation of Ras is accelerated by GEFs (*Bandaru et al., 2019*; *Boriack-Sjodin et al., 1998*; *Ehrhardt et al., 2002*; *Harrison et al., 2016*; *Vetter and Wittinghofer, 2001*).

The two activities associated with Ras – the enzymatic activity that results in the hydrolysis of GTP, and the signaling activity that enables GTP-bound Ras to bind to effector proteins – have opposing outcomes. The enzymatic activity switches off the signaling activity, and mutations that damage the catalytic center of Ras or that increase the rate of spontaneous nucleotide exchange lead to increased signaling activity. We use the terms 'activity' and 'activation' to refer to the signaling activity of Ras.

Ras is mutated frequently in cancers and some hyperproliferative developmental disorders (e.g. Noonan, Costello, and cardio-facio-cutaneous syndromes) (*Li et al., 2018*; *Prior et al., 2020*; *Prior et al., 2012*; *Young et al., 2009*). Data from cancer genomics show that most mutations occur at just three sites in Ras (Gly 12, Gly 13, and Gln 61), with the K-Ras isoform being mutated more frequently than H-Ras or N-Ras (*Figure 1—figure supplement 1A, B*; *Tate et al., 2019*). We refer to these three residues as the 'cancer hotspot' sites. In previous work, we used single-site saturation mutagenesis to assess the mutational-fitness landscape of the G-domain of H-Ras (H-Ras$^{2-166}$) by employing a high-throughput bacterial two-hybrid assay (*Bandaru et al., 2017*). The assay couples the transcription of an antibiotic-resistance factor to the binding of Ras to the RBD of C-Raf, and reports on the signaling activity of Ras (*Figure 1C*). We expected mutations at Gly 12, Gly 13, and Gln 61 to be more activating than other substitutions, reflecting the frequencies of mutations observed in cancer (*Figure 1—figure supplement 1A*). To our surprise, the bacterial screens identified multiple signal-activating mutations in Ras, with relative enrichment scores similar to those for mutations at the three cancer hotspots.

Several factors could potentially explain the difference in the mutational spectrum of Ras in cancer compared to that observed in the bacterial assay. First, the bacterial assay does not provide the complete biological context for Ras function. Ras is not membrane-localized in this system, and it is divorced from its normal complement of effector proteins. The distribution of cancer mutations reflects the mutagenic ability of specific carcinogens and the context-specific effects of Ras isoforms in different mammalian cells (*Cook et al., 2021*; *Li et al., 2018*; *Prior et al., 2020*). Analysis of the context-specific effects led to the proposal that there are 'sweet spots' at the intersection of these properties that are selected for in different cancers (*Li et al., 2018*). Nevertheless, the difference between the very narrow mutational profile of Ras in cancer and the much broader range of mutations that activate Ras in our previous deep-mutational scans is striking, and it motivated us to examine whether saturation mutagenesis could identify factors that may account for this discrepancy.

To address concerns that the broad spectrum of activating mutations seen in the bacterial assay might be artifacts of this bacterial system, we carried out saturation mutagenesis of full-length H-Ras (H-Ras$^{1-188}$) expressed in mammalian Ba/F3 cells (*Figure 1D*). The murine Ba/F3 hematopoietic cell line is dependent on the cytokine interleukin-3 (IL-3) for growth, but the IL-3 dependence can be bypassed by the expression of activated variants of tyrosine kinases, such as BCR-Abl (*Daley and Baltimore, 1988*; *Mandanas et al., 1993*; *Warmuth et al., 2007*). The Ba/F3 system is a robust assay for screening activating mutations in tyrosine kinases (*Hoover et al., 2001*; *Lee and Shah, 2015*; *Watanabe-Smith et al., 2017*). Cytokine independence can also be conferred on Ba/F3 cells by activated mutants of Ras (*Awad, 2021*; *Hoover et al., 2001*; *White et al., 2016*), providing the basis for our Ras saturation-mutagenesis screens.

We compared the mutational-fitness landscape of full-length H-Ras$^{1-188}$ in Ba/F3 cells to data obtained from bacterial saturation-mutagenesis screens done for H-Ras in the absence of GAP and GEF regulators ('unregulated Ras'), in the presence of both a GAP and a GEF ('Ras+GAP+GEF'), and

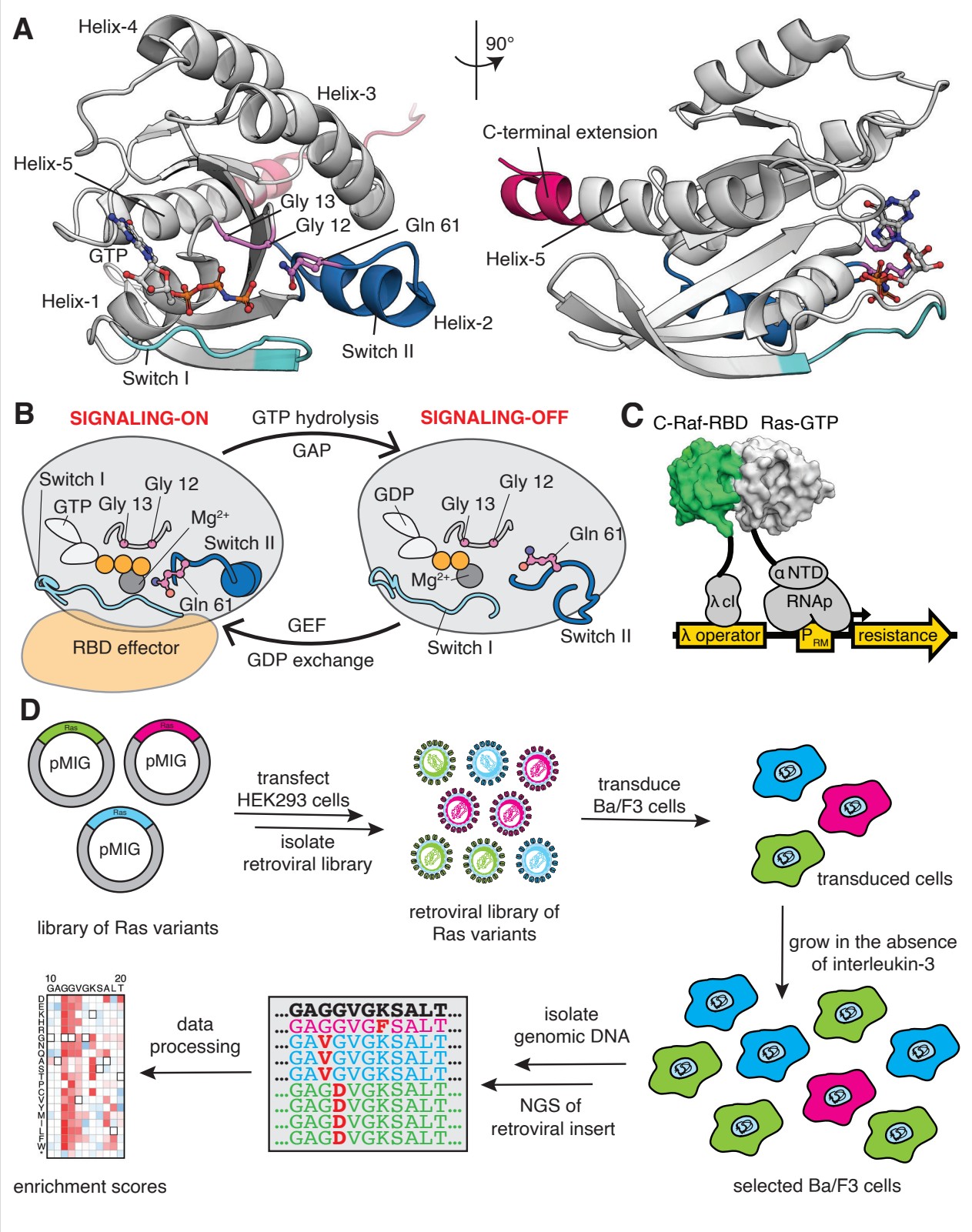

**Figure 1.** Ras G-domain, the switching cycle, and schematics of the two selection assays. (**A**) The three principal sites of cancer mutations in Ras, referred to as the three cancer hotspots – Gly 12, Gly 13, and Gln 61 – are shown in the K-Ras structure. The C-terminal helix extension is indicated. PDB ID: 2MSD (*Mazhab-Jafari et al., 2015*). (**B**) Ras cycles between signaling-active GTP-bound and signaling-inactive GDP-bound states. (**C**) The bacterial-two-hybrid system couples the C-Raf-RBD•Ras-GTP interaction to the transcription of an antibiotic resistance gene (*Bandaru et al., 2017*). Ras is fused

*Figure 1 continued on next page*

*Figure 1 continued*

to the N-terminal domain of the α-subunit of the *E. coli* RNA polymerase. C-Raf-RBD is fused to the λ-cI protein. The GAP and the GEF can be co-expressed in the system. *E. coli* cells are transformed with a DNA library of 'unselected' variants. The bacteria are grown in the presence of an antibiotic for 9 hr, then the DNA library of 'selected' variants is isolated. Next-generation sequencing (NGS) is used to count the frequency of each variant in the unselected and selected samples. (**D**) Ba/F3 assay for Ras activity. Mutant H-Ras libraries are transfected into HEK 293T cells to generate a retroviral library of mutants. Ba/F3 cells are transduced with the retroviral library. After 24 hours, a fraction of the cells are used as the unselected population, and the remainder of the cells – the selected population – are cultured for 7 days in the absence of IL-3 before harvesting them by centrifugation. The genomic DNA of the selected and unselected populations is isolated and sequenced using NGS. The relative enrichment scores are calculated using the selected and unselected counts (see *Equation 1*).

The online version of this article includes the following figure supplement(s) for figure 1:

**Figure supplement 1.** Analysis of the distribution of Ras mutations in COSMIC v94.

in the presence of a GAP alone ('Ras+GAP'). While the first two conditions yield similar patterns of mutational sensitivity, the Ras+GAP condition leads to starkly different results. The GAP switches off Ras signaling activity, and only mutations that enable Ras to evade GAP control allow cell proliferation. The mutational profile of H-Ras in the mammalian Ba/F3 cell line closely resembles the results of the bacterial screens using H-Ras co-expressed with a GAP. These results point to the importance of GAP surveillance in controlling Ras activity in Ba/F3 cells, and also demonstrate that the bacterial assay is a reliable indicator of the effects of mutations on Ras function. Strikingly, just like in the bacterial assay, the mammalian experiments reveal additional gain-of-function mutations with similar levels of activation as mutations at the cancer hotspots.

The importance of a special feature associated with the cancer hotspots emerged from the comparison of mutational profiles for longer and shorter H-Ras constructs using the bacterial assay. The shorter construct – residues 2–166, the construct typically used in crystallographic studies of H-Ras (*Pai et al., 1990*) – corresponds to the core G-domain of H-Ras and was used in our previous saturation-mutagenesis study. The longer construct spans residues 2–180 (H-Ras$^{2-180}$), and we found that it is more stable. Our earlier study had characterized several mutations outside the hotspots that activate Ras to varying degrees, but are not prominent in cancer (*Bandaru et al., 2017*). We now report the analysis of protein dynamics for three of these mutations (H27G, L120A, and Y157Q) by hydrogen-deuterium exchange (HDX) measured by nuclear magnetic resonance (NMR). The mutants show increased HDX rates throughout the protein relative to wild-type Ras, consistent with destabilization.

The saturation-mutagenesis data show that several infrequent cancer mutations (e.g. at Val 14 or Asp 119) are deleterious in the shorter construct and become activating in the longer Ras construct. We infer that these activating mutations have a destabilizing effect that only the longer Ras construct can accommodate, allowing the increased signaling capacity to be manifested. In contrast, the effect of mutations at the three cancer hotspots is independent of the Ras construct stability. We also present the results of saturation-mutagenesis experiments for K-Ras, which yield similar mutational patterns as for H-Ras. Our analysis shows that many activating mutations impact the thermodynamic stability of Ras, and the destabilizing effect of the mutations correlates with their low observed frequency in cancer.

## Results and discussion

### Saturation-mutagenesis of H-Ras in mammalian Ba/F3 Cells

Ba/F3 cells were transduced with a retroviral library of variants of full-length human H-Ras and allowed to grow for a day in the presence of IL-3. Then, a fraction of the cells were harvested and used as the 'unselected' population. The remainder of the cells were grown for a week without IL-3 in the medium (the 'selected' population). The DNA from both cell populations was harvested and sequenced to count the occurrence of each particular variant. The effect of Ras mutations on fitness is quantified by relative enrichment scores ($\Delta E_x^i$), also referred to as fitness values (*Equation 1*; see Materials and methods):

$$\Delta E_x^i = log_{10}\left[\frac{c_i^{x,selected}}{c_i^{x,unselected}}\right] - median\left(log_{10}\left[C^{wt,selected} \oslash C^{wt,unselected}\right]\right) \tag{1}$$

The first term of *Equation 1* is the logarithm of the ratio of counts (c) of observing codons representing each amino acid $x$ at each position $i$ in the selected and unselected samples. The second term of *Equation 1* is the median of the ordered list of logarithms of the elements of the vector obtained by conducting pair-wise division, denoted $\oslash$, between the selected and unselected counts ($C^{wt,selected}$ and $C^{wt,unselected}$, respectively) for the variants that are synonymous with the wild-type (wt) allele. A Ras variant with an enrichment score of zero propagates in the assay at the same rate as the wild-type variants. Variants with scores of ±1 propagate ten-fold faster or slower than wild-type variants, respectively.

The experiment was repeated twice, once while varying residues 2–160 (the core G-domain; the remaining 29 residues were not varied), and once while varying residues 2–188. The resulting fitness values are shown in the form of a heatmap in *Figure 2A*, averaged over the two replicates (*Figure 2—figure supplement 1A*). Each entry in the matrix indicates the fitness score for substituting a particular residue in Ras with one of the 20 amino acids, with shades of red and blue indicating gain or loss-of-function relative to wild-type, respectively.

There are several positions in H-Ras at which multiple substitutions lead to a strong gain-of-function. These residues include the cancer hotspots (Gly 12, Gly 13, and Gln 61), as well as several other residues that are not as frequently found to be mutated in cancer (e.g. Val 14, Arg 68, Lys 117, and Asp 119). The fact that several amino acid substitutions at each of these sites lead to increased fitness suggests that the mutations disrupt inhibitory interactions. Positions at which mutations lead to a strong loss-of-function are sparse in the dataset. Residues in the hydrophobic core of the protein, for which mutations to polar residues are expected to decrease the stability of the protein, show little or no evidence for loss of fitness when mutated. For example, the sidechains of Leu 19, Leu 79, Val 81, and Val 114 pack together in the hydrophobic core of Ras. These residues tolerate substitutions by many polar residues with no apparent reduction in fitness with respect to wild-type Ras. Since only mutations that bypass the IL-3 dependence promote cell growth in the assay, and wild-type H-Ras does not promote growth (*Figure 2—figure supplement 1B*), a neutral mutation can barely be distinguished from a deleterious mutation.

## The mutational profile for Ras in Ba/F3 cells resembles that for Ras co-expressed with a GAP in the bacterial assay

The two properties of the Ba/F3 dataset noted above, namely strong activation by many mutations at specific sites and a general sparsity of sites where mutations lead to loss-of-function, are reminiscent of the mutational profile for H-Ras²⁻¹⁶⁶ co-expressed with a GAP (H-Ras²⁻¹⁶⁶+GAP) in the bacterial assay (*Bandaru et al., 2017*). The GAP inactivates wild-type Ras by stimulating GTP-hydrolysis. Under these conditions, the assay is not sensitive to the effects of mutations that further reduce the activity of Ras by destabilizing the protein. Mutations that disrupt the interaction with the GAP, such as substitutions of Gly 12 or Gly 13, or that compromise the catalytic activity of Ras, such as substitutions of Gln 61, are strongly activating.

We used receiver operating characteristic (ROC) curves to make a quantitative determination of which of the three different bacterial experiments [H-Ras²⁻¹⁶⁶+GAP (*Figure 2A*), H-Ras²⁻¹⁶⁶+GAP+GEF (*Bandaru et al., 2017*), or unregulated H-Ras²⁻¹⁶⁶ (*Figure 3A*)] best matches the results of the Ba/F3 screen (*Figure 2B*). To generate a ROC curve, the fitness data for individual mutations in a particular bacterial dataset (e.g. H-Ras²⁻¹⁶⁶+GAP) are used to predict the fitness of mutations in the Ba/F3 experiment. A variable threshold value of fitness is used, and for each threshold value, mutations in the bacterial dataset with a fitness value greater than that threshold are considered to predict activation in Ba/F3 data. Mutations with a fitness score greater than 1.5 times the standard deviation in the Ba/F3 dataset are considered activating (i.e. true positives). For each of the bacterial datasets, an ROC curve is generated by graphing the fraction of true positives versus the fraction of false positives at various threshold settings, and the estimated area under the curve (AUC) gives a measure of the overall prediction accuracy. For a perfect correlation between the bacterial data and the Ba/F3 data, the AUC would be 1.0. The analysis shows that the Ras+GAP dataset most accurately predicts the H-Ras¹⁻¹⁸⁸ in Ba/F3 cells dataset. The AUC is 0.67 for the unregulated H-Ras²⁻¹⁶⁶ dataset (*Figure 3—figure supplement 1*) and 0.63 for H-Ras²⁻¹⁶⁶+GAP+GEF (*Figure 2—figure supplement 2B*). For H-Ras²⁻¹⁶⁶+GAP, the AUC is substantially higher at 0.84 (*Figure 2B*). Given that Ras is predominantly GDP-bound *in vivo* (*Zhao et al., 2020*), it is logical that the bacterial Ras+GAP

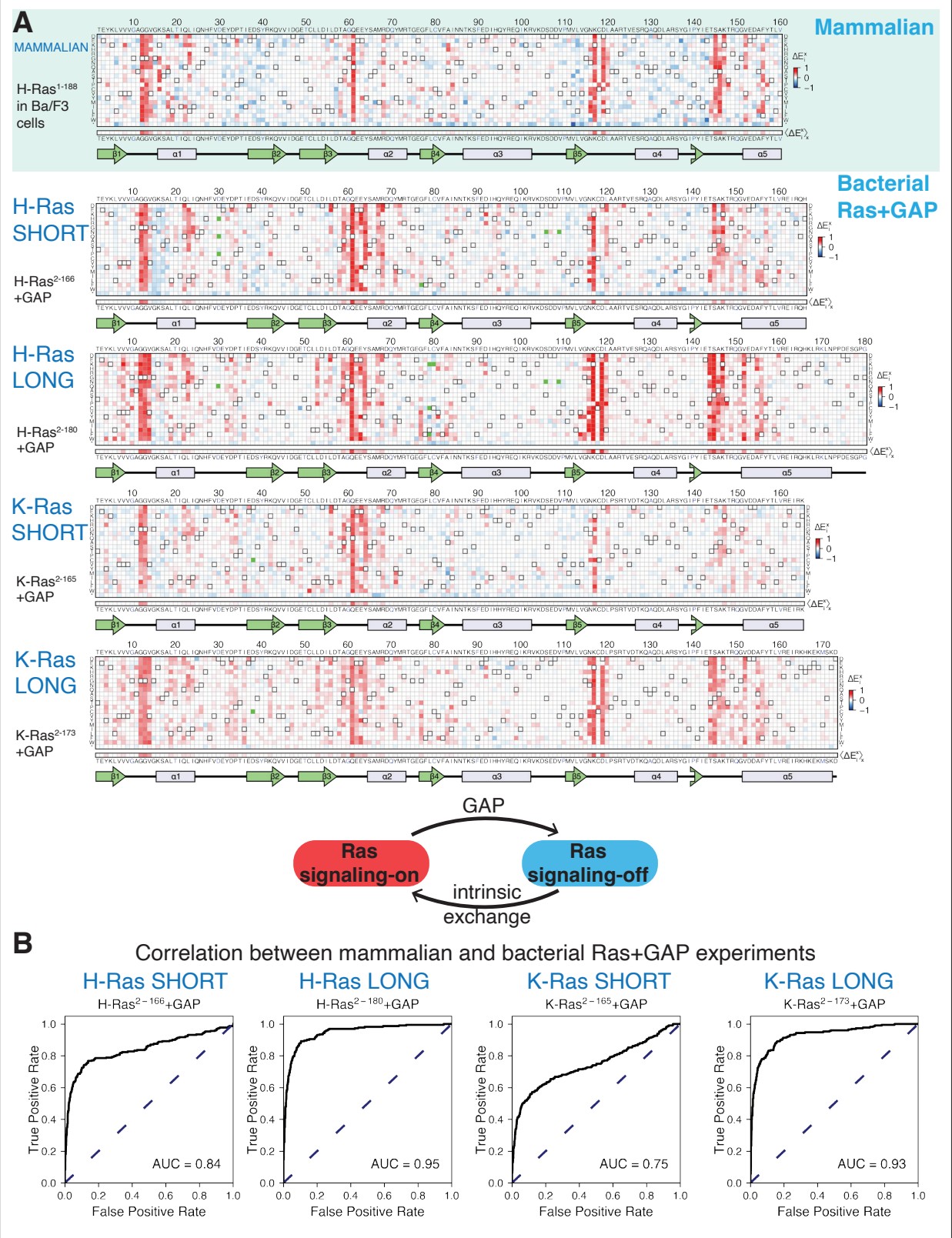

**Figure 2.** Mutational tolerance of Ras in mammalian Ba/F3 cells and the bacterial Ras+GAP experiment. (**A**) The fitness data from saturation-mutagenesis experiments are shown in the form of a matrix, where each row of the matrix represents one of the twenty natural amino acids, and each column displays a residue of the protein (*Bandaru et al., 2017*). Each entry in the matrix represents, in color-coded form, the relative enrichment score ($\Delta E_x^i$) for the corresponding variant (see *Equation 1*). The data are normalized using the distribution of enrichment scores of all the synonymous wild-

*Figure 2 continued on next page*

*Figure 2 continued*

type sequences, so the median of the distribution has a value of zero. Shades of red and blue indicate gain and loss-of-function, respectively, relative to wild-type. Green indicates variants that were not represented in the library. Stop codons are labeled as '*', and the bottom strip displays the functional effect of all amino acid substitutions at each position ($\left\langle \Delta E_x^i \right\rangle_x$) – the average taken over each column. The relative enrichment values are provided in the GitHub repository. The secondary structural elements of Ras are displayed below each matrix. The top heatmap shows the data for saturation mutagenesis of H-Ras[1-188] in mammalian Ba/F3 cells. Only the enrichment scores for variants within residues 2 and 160 are displayed in the heatmap, calculated as the mean of two biological replicates. The next four heatmaps show the data for H-Ras[2-166], H-Ras[2-180], K-Ras[2-165], and K-Ras[2-173] in the bacterial Ras+GAP experiments. The enrichment values shown are the mean of two, four, three, and four biological replicates, respectively. (**B**) The area under the curve (AUC) of receiver operating characteristic (ROC) graphs is used to determine which of the four Ras+GAP experiments better predicts the enrichment scores of mutations in the mammalian Ba/F3 cell experiment.

The online version of this article includes the following figure supplement(s) for figure 2:

**Figure supplement 1.** Mammalian Ba/F3 cell screen replicates and growth curves of wild-type H-Ras versus the G12V mutant.

**Figure supplement 2.** Mutational tolerance of Ras in the bacterial Ras+GAP+GEF experiment.

experiment – where the GAP promotes a GDP-bound state – closely resembles the mammalian Ba/F3 cell experiment.

## Extension of the C-terminal helix in Ras improves the correlation of bacterial mutagenesis data with the Ba/F3 dataset

The Ba/F3 experiments use full-length H-Ras, including the lipid-modified C-terminal hypervariable region that confers membrane anchorage. The bacterial experiments use a construct of H-Ras that spans residues 2–166, corresponding to the core G domain of H-Ras (*Pai et al., 1990*). This construct lacks 22 C-terminal residues, including the CAAX motif, where post-translational modifications occur. Membrane-anchorage of Ras is not possible within the context of the bacterial assay because the two-hybrid readout of Ras activity relies on the interaction between fusion proteins containing Ras and Raf-RBD bound to DNA.

We tested the effects of extending the C-terminal boundary of the Ras construct by conducting saturation-mutagenesis experiments with the bacterial assay using an H-Ras construct spanning residues 2–180 (*Figure 2A*). Based on the crystal structure of the K-Ras isoform (*Cruz-Migoni et al., 2019*), we expect that extending the H-Ras construct in this way lengthens the C-terminal helix of H-Ras by two helical turns or more (*Figure 1A*). The H-Ras[2-180]+GAP dataset improves the ROC AUC score for predicting the Ba/F3 data to 0.95, compared to 0.84 for the shorter construct (*Figure 2B*). There are specific sites for which mutations in the bacterial H-Ras[2-166]+GAP experiment demonstrate a different activation profile than the H-Ras[1-188] in Ba/F3 cells experiment (*Figure 2A*). For example, most mutations of Asp 119 are activating in Ba/F3 cells and H-Ras[2-180]+GAP, but these mutations are neutral or slightly inactivating in the bacterial assay with shorter H-Ras[2-166]+GAP. Mutations at the cancer hotspots are activating in all three datasets (*Figure 2A* and B).

We also performed saturation mutagenesis of K-Ras in the presence of a GAP in bacteria using two constructs, a shorter one spanning residues 2–165 (K-Ras[2-165]) and a longer one spanning residues 2–173 (K-Ras[2-173]) (*Figure 2A*), corresponding to a construct used in recent crystallographic studies (*Cruz-Migoni et al., 2019*). The K-Ras datasets are similar to the H-Ras datasets (*Figure 2A*). The ROC AUC for predicting the Ba/F3 data for shorter K-Ras[2-165]+GAP is 0.75, and the AUC increases to 0.93 for longer K-Ras[2-173]+GAP (*Figure 2B*). These data indicate that K-Ras behaves similarly to H-Ras regarding mutational tolerance. This finding is clinically relevant because the ClinVar review panel categorizes new Ras variants found in genomic studies as disease-causing or not, based on the assumption that the pathogenicity of analogous variants in the H-Ras and K-Ras genes are correlated (*Landrum et al., 2018*).

We carried out saturation-mutagenesis screens in bacteria for unregulated H-Ras and K-Ras (i.e. no co-expression of GAP and GEF) for the longer and shorter constructs (*Figure 3A*). Wild-type Ras generates a signal in the absence of the GAP (*Coyle and Lim, 2016*), and substitution of residues in the hydrophobic core by polar residues results in measurable decreases in signaling, in contrast to the Ras+GAP experiments. The data show that the longer H-Ras[2-180] and K-Ras[2-173] constructs are less sensitive to mutations of residues in the hydrophobic core than the shorter constructs (e.g. see mutations of Phe 82, *Figure 3B–C*). We also conducted screens for K-Ras in the presence of a GAP

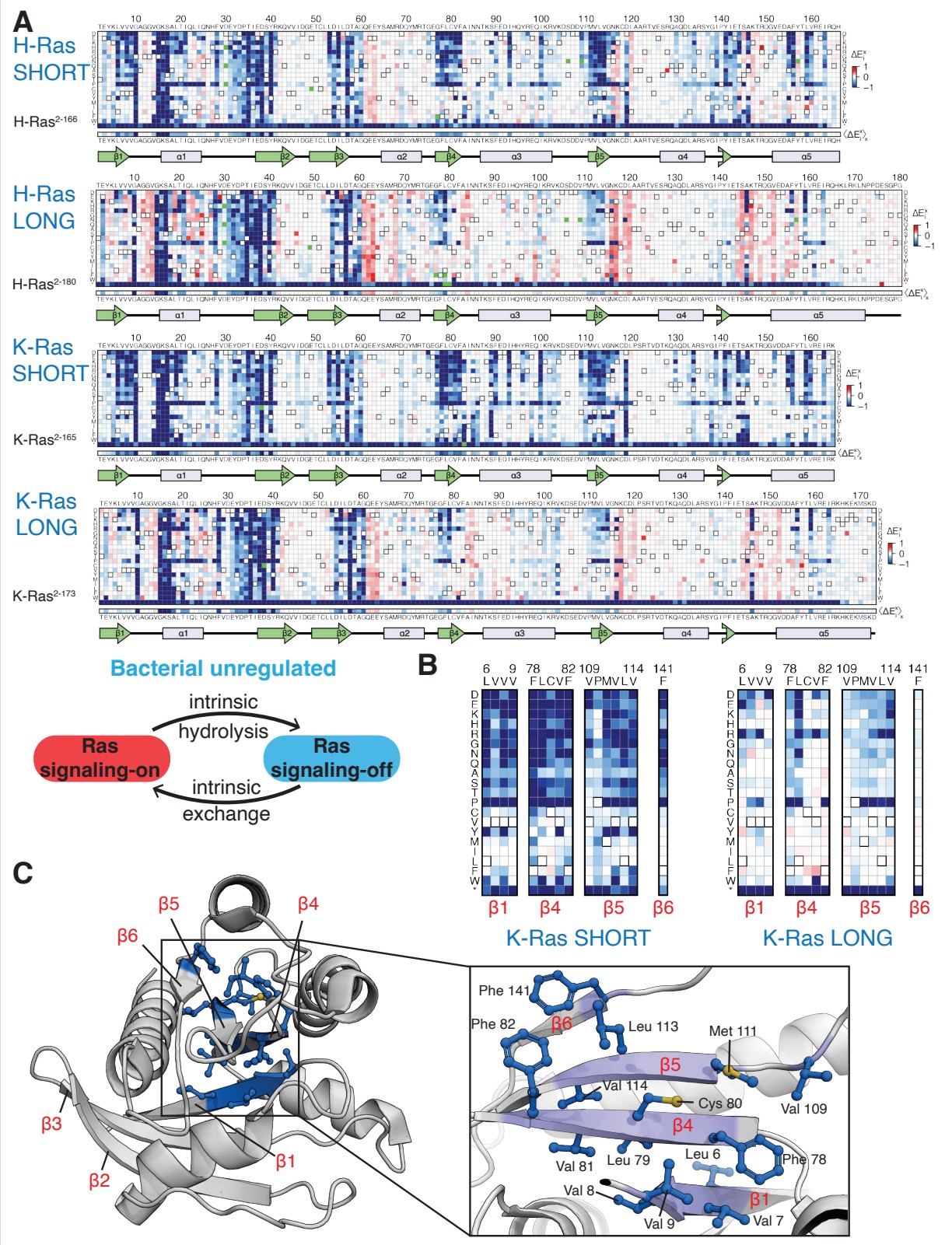

**Figure 3.** Mutational tolerance of Ras long and short constructs in the unregulated bacterial experiment. (**A**) H-Ras$^{2-166}$, H-Ras$^{2-180}$, K-Ras$^{2-165}$, and K-Ras$^{2-173}$ in the unregulated experiment. The relative enrichment values ($\Delta E_x^i$) shown are the mean of three, four, two, and four biological replicates, respectively. Stop codons are labeled as '*', and the bottom strip displays the functional effect of all amino acid substitutions at each position ($< \Delta E_x^i >_x$) – the average taken over each column. (**B–C**) Comparison of the mutational tolerance of residues in the hydrophobic core of K-Ras in the shorter and longer

*Figure 3 continued on next page*

*Figure 3 continued*

constructs. The enrichment scores for the hydrophobic core residues are shown for the two K-Ras constructs in B, and mapped onto K-Ras structure in C. PDB ID: 4OBE (*Hunter et al., 2015*).

The online version of this article includes the following figure supplement(s) for figure 3:

**Figure supplement 1.** ROC analysis between the mammalian Ba/F3 cell experiment and the unregulated bacterial H-Ras and K-Ras experiments.

and a GEF, and the mutational landscape presents the same patterns described for the unregulated experiments (*Figure 2—figure supplement 2A*).

## Ras mutations and construct length impact fold stability

We measured the stability of two H-Ras constructs bound to GDP, one corresponding to the core G-domain (residues 1–166), and one in which the C-terminal end of the construct is extended by seven residues (residues 1–173) using two assays with purified Ras proteins. In the first assay, we carried out a urea titration and monitored the equilibrium unfolding transition by following the circular dichroism (CD) signal at 222 nm. In the second assay, we also monitored the urea-induced unfolding using pulse proteolysis. Under pulse conditions, all the unfolded protein is cleaved, and the native protein remains undigested (*Kim et al., 2009*; *Park and Marquesee, 2005*; *Samelson et al., 2016*). The fraction of folded protein can be estimated by measuring the intensity of the band corresponding to full-length Ras on an SDS/PAGE gel. The intensities of the bands at each urea concentration are then fit to a two-state denaturation model, with the urea-dependence (the m-values determined by CD) used to determine midpoint urea concentration (Cm).

The stabilities ($\Delta G_{unf}$) determined by pulse proteolysis are in close agreement with those determined by CD. When monitored by CD, the values of $\Delta G_{unf}$ for the shorter and longer H-Ras constructs are 22.2 ± 1.6 kJ.mol$^{-1}$ and 29.9 ± 1.4 kJ.mol$^{-1}$, respectively (*Figure 4A*). When measured by pulse proteolysis, the shorter and longer constructs have values of $\Delta G_{unf}$ of 24.3 ± 1.3 kJ.mol$^{-1}$ and 31.1 ± 1.5 kJ.mol$^{-1}$, respectively (*Figure 4A*). Both assays indicate that truncating the C-terminal helix destabilizes H-Ras by ~7 kJ.mol$^{-1}$. This effect is consistent with the known ability of terminal-helix stabilization to help maintain the entire protein fold (*Rosemond et al., 2018*).

Using the pulse proteolysis assay, we measured the stability of three cancer hotspot mutants (Q61L, G12V, G13D), and two infrequent – or absent – mutants (K117N, and D119A) (*Figure 4B*). The stabilities of the Q61L and G12V variants of H-Ras[1-166] are indistinguishable from that of wild-type ($\Delta G_{unf}$ of 28.3 ± 4.3 kJ.mol$^{-1}$ for G12V and 23.9 ± 1.0 kJ.mol$^{-1}$ for Q61L). For the G13D, K117N, and D119A variants of H-Ras[1-173], the values of $\Delta G_{unf}$ are 22.1 ± 1.5 kJ.mol$^{-1}$, 6.8 ± 0.5 kJ.mol$^{-1}$, and 6.2 ± 0.8 kJ.mol$^{-1}$, respectively. G13D destabilizes Ras by a moderate amount, ~ 10 kJ.mol$^{-1}$, comparable to the effect of shortening the C-terminal tail. The K117N and D119A mutations destabilized Ras much more substantially, by ~25 kJ.mol$^{-1}$.

Mutations at Lys 117 and Asp 119 as well as mutations at other sites (e.g. Val 14, Phe 28, Leu 120, Ala 146, Lys 147, and Phe 156), increase the rate of intrinsic nucleotide release (*Baker et al., 2013*; *Bandaru et al., 2017*; *Bera et al., 2019*; *Cirstea et al., 2013*; *Cool et al., 1999*; *Gelb and Tartaglia, 2006*; *Poulin et al., 2019*; *Quilliam et al., 1995*; *Reinstein et al., 1991*). Furthermore, two such mutations – V14I and A146T – have been shown to facilitate nucleotide exchange by destabilization and opening of the GTP-binding site (*Bera et al., 2019*; *Poulin et al., 2019*). Hence, stability and nucleotide affinity in Ras are coupled, consistent with Ras stability being dependent on nucleotide binding (*Zhang and Matthews, 1998a*; *Zhang and Matthews, 1998b*).

We use the term 'stability-dependent gain-of-function' mutations to denote the subset of mutations that switch from an activating phenotype in the longer constructs to a neutral or deleterious phenotype in the shorter constructs or vice versa. We hypothesize that the same mutation may activate the longer Ras construct due to the compensating stability provided by the extra residues, but in the shorter and less stable construct, the decrease in stability due to the mutation leads to reduced levels of folded Ras. There are also a few mildly destabilizing mutations that display a stronger activating phenotype in the shorter Ras construct than in the longer one (e.g. some mutations at Tyr 157). In such cases, we speculate that the mild destabilization only leads to an observable phenotype in the context of the shorter, less-stable, Ras construct. We do not consider G13D a stability-dependent gain-of-function mutation even though it increases nucleotide exchange (*Hunter et al., 2015*; *Smith*

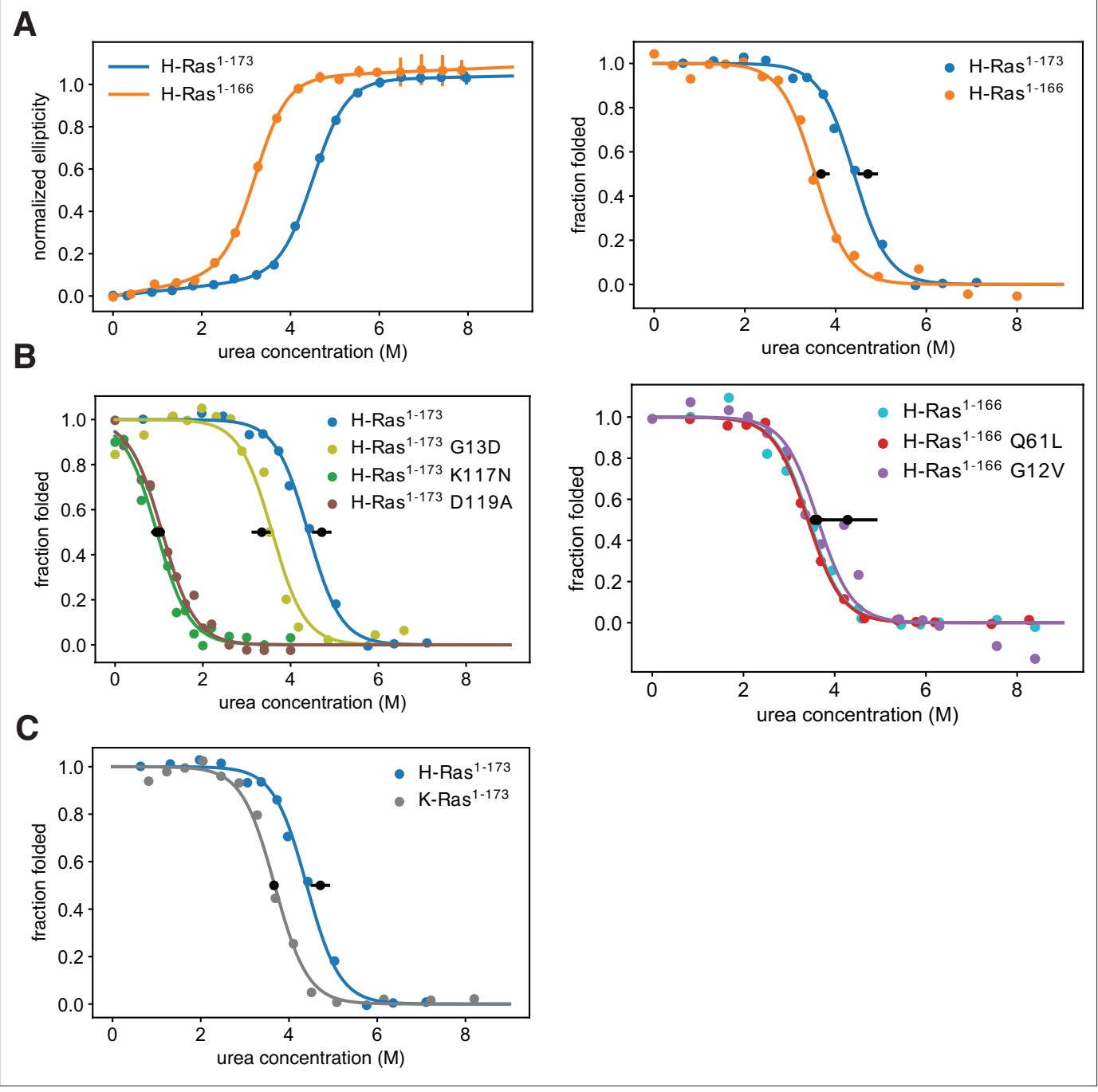

**Figure 4.** Thermodynamic stability measurements. (**A**) The shorter H-Ras[1-166] construct has an unfolding free-energy change ($\Delta G_{unf}$) of 22.2 ± 1.6 kJ·mol[-1], and the longer H-Ras[1-173] construct has a $\Delta G_{unf}$ of 29.9 ± 1.4 kJ·mol[-1] when measured by circular dichroism (left). When measured by pulse proteolysis (right), the shorter construct has a $\Delta G_{unf}$ of 24.3 ± 1.3 kJ·mol[-1], and the longer construct has a $\Delta G_{unf}$ of 31.1 ± 1.5 kJ·mol[-1]. Both assays indicate that truncating the C-terminal helix destabilizes H-Ras by ~7 kJ·mol[-1]. The CD measurements were conducted at 25 °C, using 35.5 µM GDP and 0.5 µM Ras. CD signal error bars are ± two times the standard deviation of four 15 second reads. (**B**) Pulse proteolysis measurements of three cancer hotspots mutants (G12V, G13D, Q61L) and two stability-dependent gain-of-function mutants (K117N, D119A). The $\Delta G_{unf}$ for G13D, K117N, and D119A mutants in the H-Ras[1-173] construct are 22.1 ± 1.5 kJ·mol[-1], 6.8 ± 0.5 kJ·mol[-1], and 6.2 ± 0.8 kJ·mol[-1], respectively. The G12V and Q61L variants were studied in the H-Ras[1-166] construct. The $\Delta G_{unf}$ for G12V, Q61L, and new replicates of wild-type are 28.3 ± 4.3 kJ·mol[-1], 23.9 ± 1.0 kJ·mol[-1], and 23.5 ± 1.0 kJ·mol[-1], respectively. Hence, these cancer hotspot mutations are not destabilizing. (**C**) Pulse proteolysis measurements of H-Ras[1-173] and K-Ras[1-173]. K-Ras[1-173] has a $\Delta G_{unf}$ of 24.2 ± 0.1 kJ·mol[-1]; hence, H-Ras[1-173] is more stable than K-Ras[1-173] by ~7 kJ·mol[-1]. The proteolysis experiments in A-C were conducted at 25 °C, using 100 µM GDP and 1.6 µM Ras. Midpoint marker error bars are ± the standard deviation of the estimated midpoint concentration ($\Delta G_{unf}/m$) across pulse proteolysis replicates.

et al., 2013) and destabilizes Ras ($\Delta G_{unf}$ of 22.1 ± 1.5 kJ.mol$^{-1}$). H-Ras$^{1-173}$ G13D can buffer the ~7 kJ. mol$^{-1}$ destabilizing effect of the C-terminal helix truncation, and that is why G13D is activating in both shorter and longer Ras+GAP experiments.

We compared the stability of longer constructs of the two Ras isoforms: H-Ras$^{1-173}$ and K-Ras$^{1-173}$. Pulse proteolysis analyses indicate that H-Ras$^{1-173}$ is more stable than K-Ras$^{1-173}$ by ~7 kJ.mol$^{-1}$ ($\Delta G_{unf}$ of 24.2 ± 0.1 kJ.mol$^{-1}$ for K-Ras$^{1-173}$, *Figure 4C*). In the bacterial saturation-mutagenesis assays, activating mutations in all the H-Ras datasets have higher enrichment values on average than activating mutations in the corresponding assays with K-Ras (*Figures 2A and 3A*). We hypothesize that the lower stability of K-Ras reduces the activating effect of many mutations.

## Mutations that activate Ras by increasing nucleotide exchange result in decreased conformational stability as measured by HDX

In our earlier study of the mutational fitness landscape of Ras, we identified several activating mutations that increased nucleotide exchange or decreased the hydrolysis rate (*Bandaru et al., 2017*), but are either not found, or are not prominent, in cancer. We selected four such mutations in H-Ras$^{1-166}$ for analysis of protein dynamics by HDX: H27G, Q99A, L120A, and Y157Q (*Figure 5B*). Of the four analyzed mutants, L120A, H27G, and Y157Q are mildly activating in the unregulated and Ras+GAP experiments. Our new data, which are statistically more robust than those the previous analysis was based on, indicate that the Q99A mutation is neutral, or nearly so, rather than activating.

We measured amide HDX rates by NMR to probe the local conformational stability of the H-Ras variants in solution. $^1$H-$^{15}$N Heteronuclear Single Quantum Coherence (HSQC) spectra were used to follow the exchange of backbone amides from hydrogen to deuterium over a 24-hr period (*Figure 5A*). In these experiments, assignments were based on previously obtained assignments of wild-type Ras bound to GMP-PNP, which were confirmed using an HNCA spectrum (*O'Connor and Kovrigin, 2012*). For the HDX analysis, an HSQC spectrum was recorded at each timepoint, and the volumes under all observable and assigned peaks were integrated. For each peak, the decay in the integrated peak volume with exchange time was fit to a single exponential (e.g. Lys 147 peak, see *Figure 5C*). In this way, the local hydrogen-exchange behavior was determined on a residue-by-residue basis. We used a conservative approach in which peaks where the decay process did not fit to a single exponential with $R^2 > 0.7$ were excluded from the analysis. The time constants determined by this procedure were then used to compare the local conformational stability of each observable residue between the wild-type protein and the mutant proteins.

The exchange behavior of the residues (EX1 vs. EX2) was confirmed by HDX by mass spectrometry (MS) (*Figure 5—figure supplement 3*). When HDX is monitored by MS, EX1 behavior results in two peaks (so-called bimodal behavior). In contrast, EX2 behavior results in a single isotopic envelope that moves to larger m/z with time. Under EX1 conditions, the rate of labeling reports on the rate of the amide opening, and the bimodal behavior of a peptide arises from the cooperative behavior of amides next to each other. Except at the very N and C terminus of the protein, all peptides show EX2 behavior, confirming that the exchange rates measured by NMR report on the free energy of exchange and can be fit to a single exponential decay function.

The HDX rates measured for residues in the H27G, L120A, and Y157Q H-Ras variants have, on average, a twofold increase over the HDX rates of the same residues in wild-type H-Ras (*Figure 5D* and *Figure 5—figure supplement 2*). L120A and wild-type H-Ras have similar GTP-bound structures (*Bandaru et al., 2017*), yet the HDX measurements indicate that the conformational stability of the L120A structure has changed, and many residues show less protection in the presence of this mutation. This is in marked contrast to the exchange rates observed for residues in the Q99A variant, which is not activating in the new saturation-mutagenesis experiments. As demonstrated in *Figure 5D* and *Figure 5—figure supplement 2*, the HDX rates for this variant are similar to wild-type H-Ras. The fold-change in the HDX rate averages around 1.0 for Q99A versus wild-type, with a maximal change being in the negative, rather than positive, direction (slower HDX rate). For example, the HDX rate measured for Asp 119 in Q99A is five-fold lower than wild-type. Using the pulse proteolysis assay, we measured the stability of L120A and Q99A in the H-Ras$^{1-166}$ construct (*Figure 5—figure supplement 1*). L120A is destabilizing by ~10 kJ.mol$^{-1}$ ($\Delta G_{unf}$ of 14.0 ± 0.3 kJ.mol$^{-1}$), and Q99A has about the same $\Delta G_{unf}$ as wild-type ($\Delta G_{unf}$ of 23.0 ± 0.7 kJ.mol$^{-1}$). Both measured $\Delta G_{unf}$ values are consistent with the measured HDX rates.

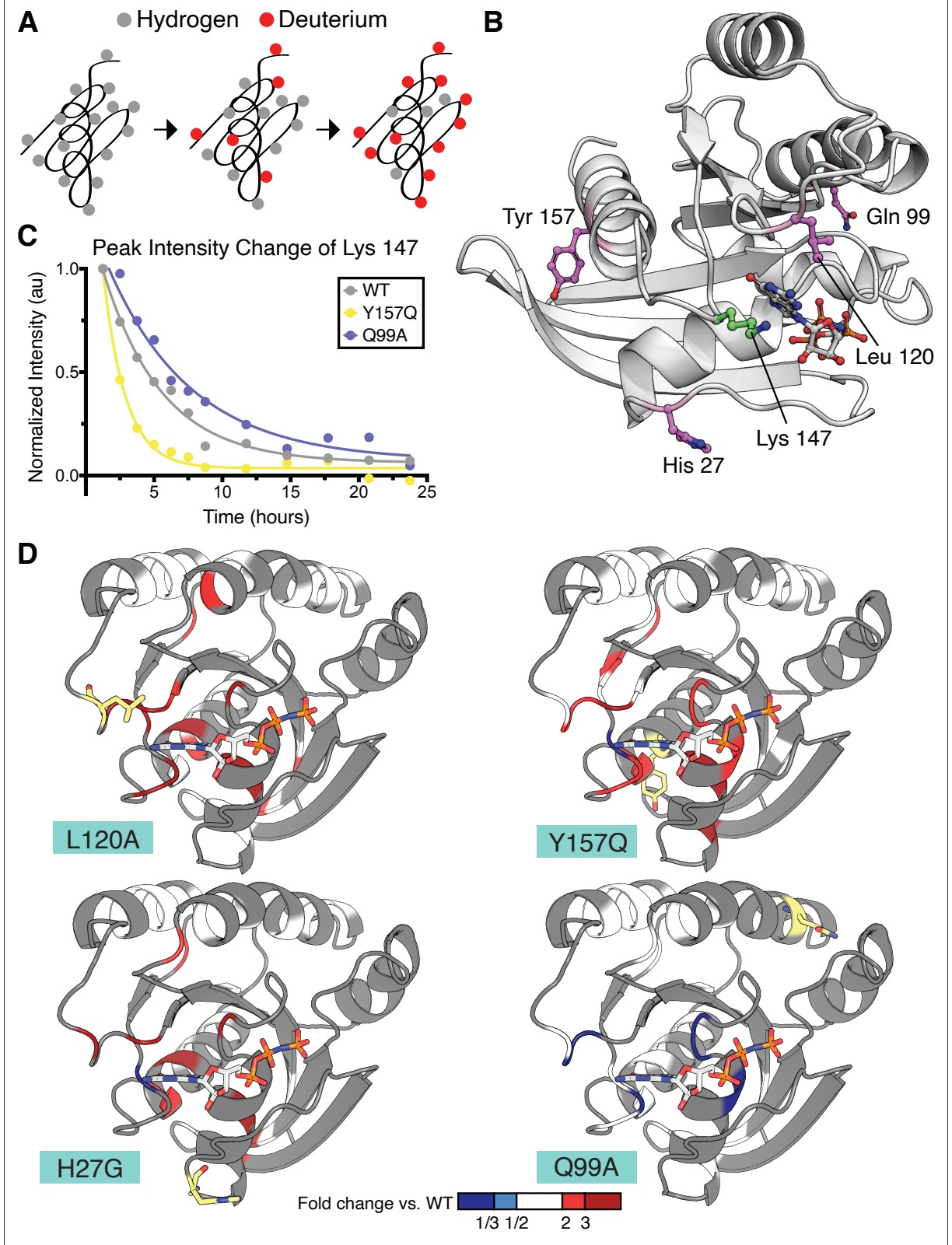

**Figure 5.** Hydrogen to deuterium exchange by NMR for activating H-Ras mutants. (**A**) Schematic of expected backbone hydrogen to deuterium exchange (HDX) over time when lyophilized protein is resolubilized in $D_2O$. (**B**) Structure displaying the four H-Ras mutants analyzed by HDX and the residue measured in C. PDB ID: 5P21 (***Pai et al., 1990***). (**C**) Integrated peak volume change over time for the Lys 147 for wild-type H-Ras[1-166] (WT), Y157Q, and Q99A samples. The line represents a single-exponential fit, from which the time-constant can be converted to exchange rate. (**D**) Fold

*Figure 5 continued on next page*

*Figure 5 continued*

change in exchange time ($k_{obs,mutant}/k_{obs,WT}$) plotted onto the WT structure of H-Ras. The mutated sites are highlighted in yellow. Backbone is colored at positions where both WT and mutant exchange could be measured in order to determine fold change. This fold change is represented on the scale reflected at the bottom of the panel. Dark blue color represents a mutant exchange less than three times slower than WT, light blue color between a two and three times slower, white between two times slower and two times faster, light red color between two and three times faster, and dark red color greater than three times faster.

The online version of this article includes the following figure supplement(s) for figure 5:

**Figure supplement 1.** Thermodynamic stability measurements of L120A and Q99A.

**Figure supplement 2.** Relative change in hydrogen to deuterium exchange for each mutant.

**Figure supplement 3.** Wild-type H-Ras exhibits EX1 behavior at the N and C terminus.

The largest changes in protection in the L120A, H27G, and Y157Q variants occur in the residues around the nucleotide-binding pocket of Ras (*Figure 5D*). Similar results have been reported for two activating mutations – V14I and A146T – that also increase nucleotide exchange and are found in cancer at a low frequency (*Bera et al., 2019*; *Poulin et al., 2019*). Taken together, our pulse proteolysis data and the HDX measurements indicate that many mutations in Ras achieve activation at the expense of destabilization and subsequent GDP release. Hence, the interplay between stability and activity in Ras may contribute towards explaining the low frequency of destabilizing mutants in cancer.

## Saturation-mutagenesis datasets correctly predict the sites of cancer mutations

To compare the results of the saturation-mutagenesis experiments with Ras mutations found in cancer, we referred to the Catalogue of Somatic Mutations in Cancer (COSMIC) (*Tate et al., 2019*). Every specific mutation in any Ras isoform, for example, G12V, that appears in the COSMIC database more than five times is included in the set of cancer mutations that we analyze. There are at least 70 sites in Ras where mutations are found in the COSMIC database, and 28 of those fulfill the five-count cutoff (*Figure 6A*). Although each Ras isoform has a different mutational profile, Gly 12, Gly 13, and Gln 61 are mutated with the greatest frequency: ~ 40,000 times, ~ 8000 times, and ~6500 times, respectively. Mutations at the remaining sites are observed far less frequently, with most mutations occurring only tens of times. For example, mutations at Asp 119 occur only seven times in the database (H-Ras).

To determine the strength of the correlation between the mutations in the saturation-mutagenesis datasets and the mutations in the cancer database, we determined ROC curves to predict cancer mutants based on the mutagenesis datasets (*Figure 6B* and *Figure 6—figure supplement 1*). This analysis ignores the relative frequencies of mutations in the COSMIC database. For a given threshold value of fitness, we consider a mutation to be activating if its fitness value is greater than the threshold. If an activating mutation is present in the filtered set of COSMIC database mutations, we regard the prediction to be a true positive, and if it is not present in the filtered set, it is scored as a false positive. Variants that are also present in an alternate database, gnomAD (*Karczewski et al., 2020*), which lists non-pathogenic mutations, are used to determine the true negatives. The fraction of true and false positive mutations is calculated for different threshold values, and the fraction of true positives is graphed as a function of the fraction of false positives to yield the ROC curve.

For predicting cancer mutations, the AUC for H-Ras$^{1-188}$ in Ba/F3 cells is 0.89, the highest value among all of the mutagenesis datasets (*Figure 6B*). The AUC for the Ras+GAP bacterial datasets for H-Ras$^{2-180}$ and K-Ras$^{2-173}$ is 0.85, and for H-Ras$^{2-166}$ and K-Ras$^{2-165}$ (with GAP), it is 0.86 and 0.82, respectively (*Figure 6—figure supplement 1*). Thus, the saturation-mutagenesis datasets are quite accurate at predicting cancer mutations. For the Ba/F3 dataset, for example, a threshold value that results in an ~80% successful prediction of the cancer mutations results in a false-positive fraction of ~30%. Using the dataset with the shorter H-Ras$^{2-166}$ without a GAP or a GEF, a fitness-threshold value that yields a true positive fraction of 80% yields a false-negative fraction of ~90% (*Figure 6—figure supplement 1*). This illustrates the predictive ability of the Ba/F3 dataset or the bacterial Ras+GAP datasets compared to the unregulated datasets. We conclude that the cancer mutations gain their power by evading GAP-mediated inactivation, unleashing the intrinsic capacity of Ras to generate signaling activity.

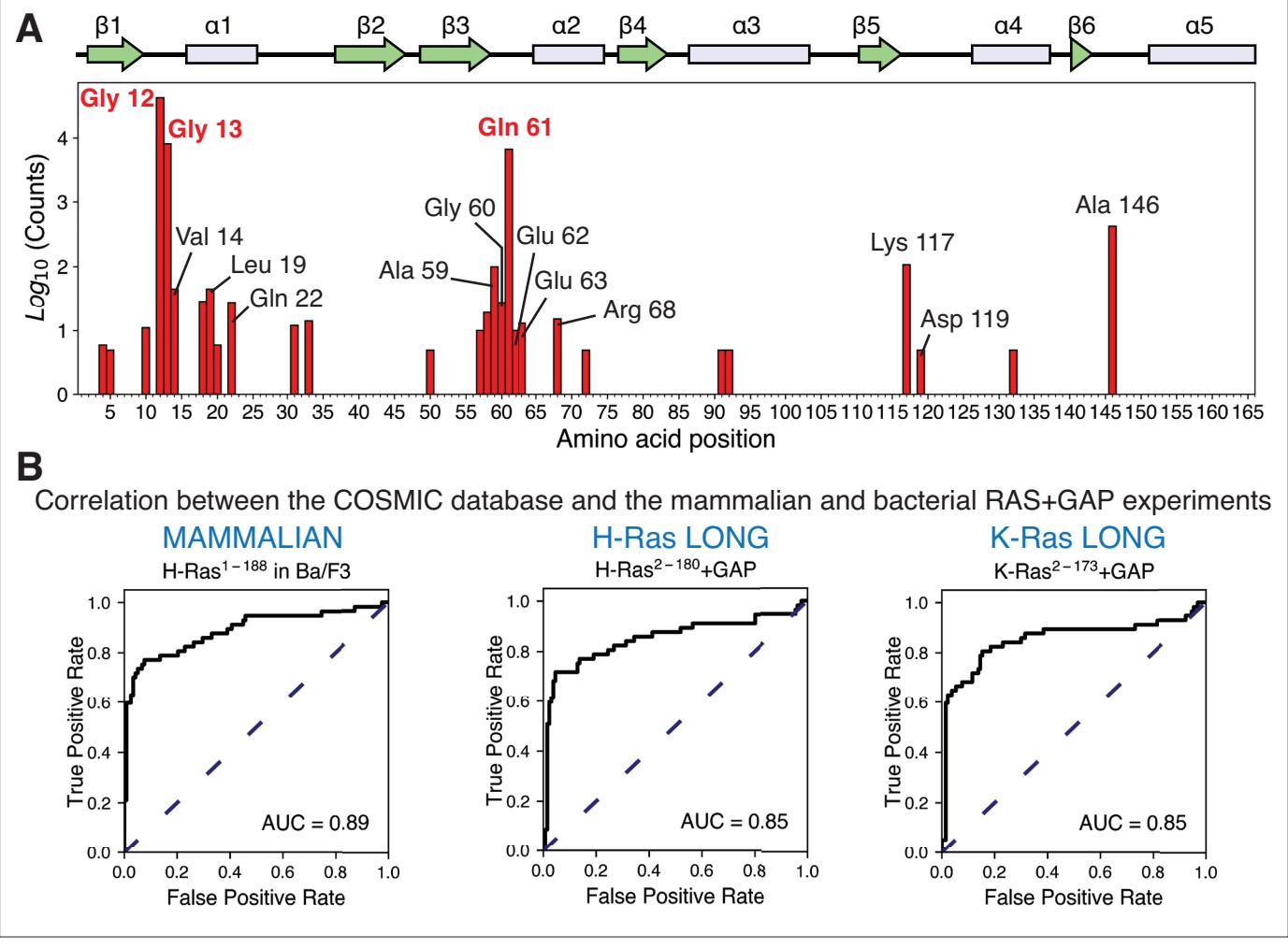

**Figure 6.** Prediction of the cancer mutations by the mammalian Ba/F3 cell and Ras+GAP experiments. (**A**) Counts per residue of Ras variants that appear in the COSMIC v94 database and pass the five-count cutoff (*Tate et al., 2019*). There are twenty-eight residues where at least one mutation is present five or more times: residues 4, 5, 10, 12, 13, 14, 18, 19, 20, 22, 31, 33, 50, 57, 58, 59, 60, 61, 62, 63, 68, 72, 91, 92, 117, 119, 132, and 146. (**B**) ROC analysis of the H-Ras$^{1-188}$ in mammalian Ba/F3 cells, H-Ras$^{2-180}$+GAP, and K-Ras$^{2-173}$+GAP datasets. For a given threshold enrichment value, we count a mutation as activating if its enrichment value is greater than the threshold. If an activating mutation is present in the COSMIC database five or more times, we count the prediction as a true positive. Mutations found in the gnomAD database are used as true negatives (*Karczewski et al., 2020*). The full list of mutations used in the analysis is provided in the Github repository.

The online version of this article includes the following figure supplement(s) for figure 6:

**Figure supplement 1.** Prediction of the cancer mutations by the bacterial experiments.

**Figure supplement 2.** Prediction of the cancer mutations using the cBioPortal dataset as true positives.

We have repeated the ROC analysis using a second set of true positives derived from the oncogenic mutations compiled in the cBioPortal database (*AACR Project GENIE Consortium, 2017*; *Cerami et al., 2012*; *Gao et al., 2013*). This database compiles data from different cancer studies from those used in the COSMIC database. While there is good agreement between the COSMIC and the cBioPortal databases for the most frequent mutations, this correlation diverges for the mutations with lower counts (*Figure 6—figure supplement 2A*). Using the cBioPortal database as the reference, the Ba/F3 and the bacterial Ras+GAP assays are still the most accurate predictors of cancer mutations (AUC values of 0.86 for the Ba/F3 dataset, 0.85 for H-Ras$^{2-180}$+GAP, 0.85 for K-Ras$^{2-173}$+GAP, 0.87 H-Ras$^{2-166}$+GAP, and 0.83 K-Ras$^{2-165}$+GAP) (*Figure 6—figure supplement 2B*).

## Stability-dependent gain-of-function mutations in Ras are less frequently found in the COSMIC database

Comparing the mutational profiles for the long and short Ras constructs in the bacterial Ras+GAP experiments (*Figure 7A*) identifies a feature that distinguishes the cancer hotspots from the less frequent sites: the activating effect of mutations at the less frequent sites, but not at the cancer hotspots, is contingent on the stability of the particular Ras construct. There are at least 24 sites in Ras where we identified activating mutations in the bacterial H-Ras+GAP mutational screens, and these are listed in *Figure 7—figure supplement 4*. We divide them into three groups: (i) the three cancer hotspots, (ii) Switch II sites, and (iii) stability-dependent gain-of-function sites.

The mutational-sensitivity profile for the three cancer hotspots is negligibly affected by the construct length in the Ras+GAP experiments (*Figure 7B and D*). Some mutations at Gly 12, Gly 13, and Gln 61 activate Ras in the saturation-mutagenesis experiments but are not observed in the COSMIC database. These mutations are all multiple-nucleotide variants (MNVs) that require more than one base-pair change. Approximately 97% of cancer mutations are single-nucleotide variants (SNVs), due to the low likelihood of multiple-nucleotide variants. For instance, the codon that translates to residue Gly 12 in wild-type Ras is GGC. The six single-nucleotide variant codons that translate to a different amino acid (AGC, CGC, TGC, GAC, GCC, and GTC) account for 99.7% of all the substitutions at Gly 12 found in the COSMIC database (*Figure 1—figure supplement 1C*).

The second group of residues is located in Switch II of Ras (Ala 59, Glu 62, Glu 63, Tyr 64, Met 67, and Arg 68). These residues show diminished sensitivity to mutations in the Ba/F3 assay compared to the bacterial assay (*Figure 7B*). The bacterial assay depends only on the interaction between the Raf-RBD and Ras, which in turn relies mainly on Switch I, rather than Switch II (*Fetics et al., 2015*). In the proper biological context, Ras GEFs (e.g. SOS and RasGRF1) utilize Switch II, as do effector proteins such as PI3K and RalGDS (*Figure 7—figure supplement 1A-D*; *Huang et al., 1998*; *Margarit et al., 2003*; *Pacold et al., 2000*; *Quilliam et al., 1996*). We surmise that Switch II mutations are less activating in the Ba/F3 dataset because disrupting Switch II function blocks the ability of Ras to signal properly, which also explains why mutations in this region are rarely seen in cancer databases. To verify this hypothesis, we measured SOS-stimulated nucleotide exchange rates for selected H-Ras mutants, using purified proteins and a fluorescent-nucleotide-release assay (*Boriack-Sjodin et al., 1998*; *Figure 7—figure supplement 1E*). Mutations of Gln 63, Tyr 64, Met 67, and a few substitutions at Tyr 71 are activating in the bacterial Ras+GAP assay but not in the Ba/F3 assay. The *in vitro* measurements show that mutation of all four residues in the Switch II region, as well as for mutations in Switch I, lead to a reduction of SOS-catalyzed nucleotide exchange rates.

The third group consists of 16 stability-dependent gain-of-function sites (Val 8, Val 14, Leu 19, Leu 53, Gly 77, Gly 115, Asn 116, Lys 117, Asp 119, Thr 144, Ser 145, Ala 146, Lys 147, Val 152, Ala 155, and Phe 156). Mutations at these sites are infrequently found in the COSMIC database, and they are more activating in the longer Ras construct than in the shorter one (*Figure 7A*). We quantified this difference (*Equation 2*) as described in a previous saturation-mutagenesis epistatic study (*McLaughlin et al., 2012*):

$$\left\langle \Delta\Delta E_x^i \right\rangle_x = \frac{1}{n} \sum_{x=1}^{n} \Delta E_{x(H-Ras^{2-166}+GAP)}^i - \Delta E_{x(H-Ras^{2-180}+GAP)}^i \tag{2}$$

The equation calculates the pair-wise difference ($\Delta\Delta E_x^i$, also named epistasis) of the enrichment of the same mutation $x$ in the H-Ras+GAP experiments of shorter and longer constructs. Then, for each position $i$ in Ras, the average epistatic effect ($\left\langle \Delta\Delta E_x^i \right\rangle_x$) of the 19 possible substitutions is calculated. Most stability-dependent gain-of-function sites cluster around the nucleotide (*Figure 7C*), and likely their mechanism of activation is to increase nucleotide exchange through destabilization.

The observation that the phenotype of mutations at the cancer hotspots remains unchanged between the longer and shorter constructs (*Figure 7A, B and D*) implies that the impact of these mutations on protein stability is mild and should not affect the mutational profile of Ras. We examined this question by fixing a mutation at one of these sites (Q61L) in K-Ras[2-173] and conducting a saturation-mutagenesis experiment with this background in unregulated and Ras+GAP conditions (*Figure 7—figure supplement 2*). Our expectation was that if the Q61L mutation led to substantial destabilization, then the mutational profile would resemble that of the shorter construct, rather than the longer construct. However, the observed loss-of-function patterns are similar to results for

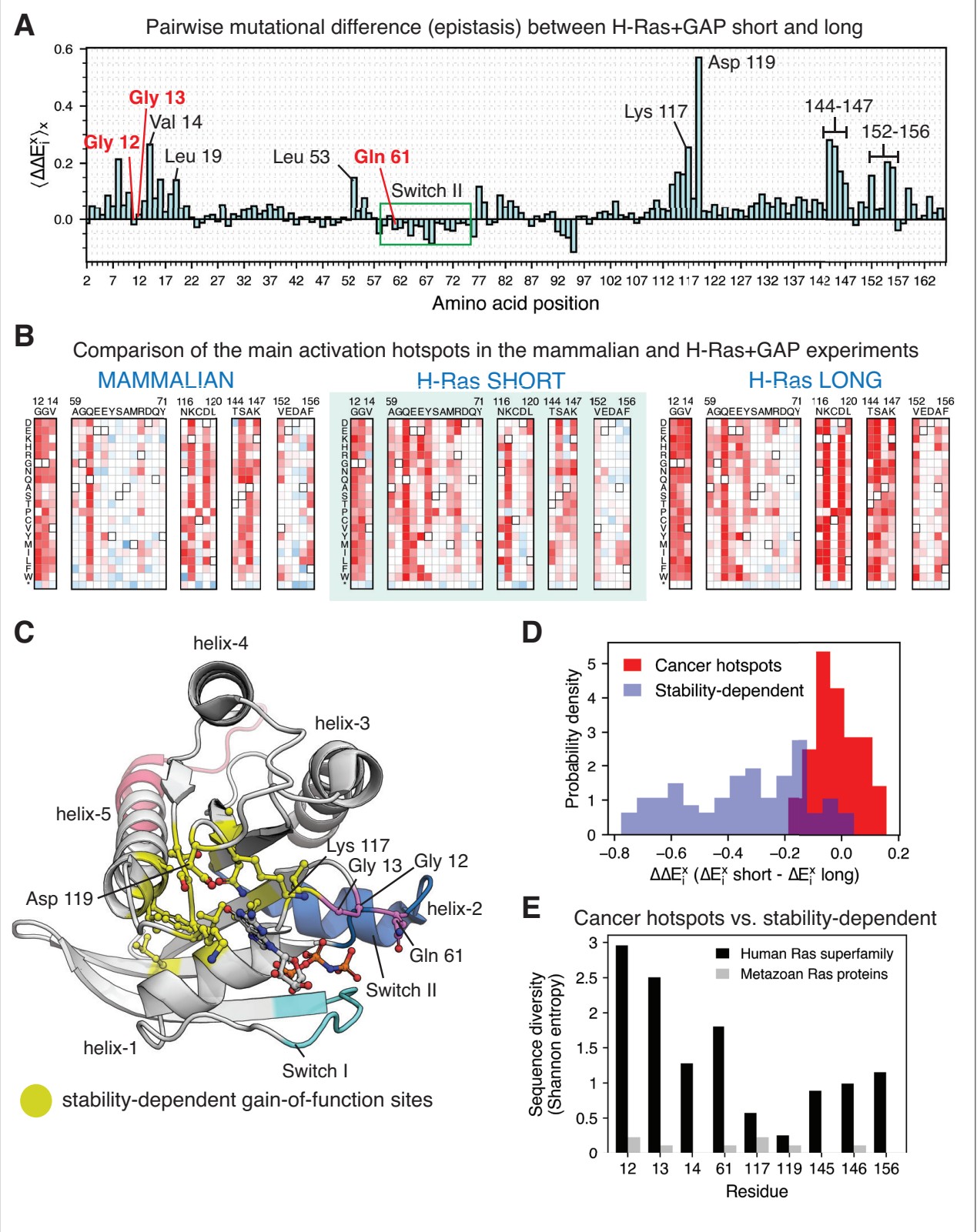

**Figure 7.** Analysis of the phenotype of activating hotspots in the screens (**A**) Comparison of the mutational sensitivity profiles for long and short H-Ras+GAP datasets. The pairwise difference (epistasis) is shown averaged over amino acids at each position ($\left\langle \Delta\Delta E_x^i \right\rangle_x$). *Figure 7—figure supplement 4* compiles the activating sites where there is epistasis. (**B**) Comparison of enrichment values for substitutions at activating sites in the H-Ras+GAP and

*Figure 7 continued on next page*

Figure 7 continued

mammalian Ba/F3 cell experiments. Substitutions at the cancer hotspots are activating, whereas substitutions at other sites (e.g.,Val 14 or Asp 119) are gain-of-function only in the longer constructs. (**C**) The stability-dependent gain-of-function sites are mapped onto Ras structure. PDB ID: 2MSD (**Mazhab-Jafari et al., 2015**). (**D**) Comparison of the pairwise difference ($\Delta\Delta E_x^i$) for activating mutations at the stability-dependent gain-of-function sites and at the cancer hotspots in long and short H-Ras+GAP datasets. Only mutations that are activating in the longer construct ( > 1.5 times the standard deviation) are considered in this analysis. (**E**) Sequence diversity analysis of the top-nine gain-of-function mutations found in the H-Ras[1-188] in mammalian Ba/F3 cells. Two separate multiple sequence alignments (MSAs) were used in the analysis: a MSA of the ~150 proteins in the human Ras superfamily (**Rojas et al., 2012**), and a MSA of the Ras ortholog sequences found in an evolutionary analysis of metazoan Ras (**Bandaru et al., 2017**).

The online version of this article includes the following figure supplement(s) for figure 7:

**Figure supplement 1.** Structures of C-Raf-RBD, PI3K, p120GAP, and SOS[cat] in complex with H-Ras, and SOS[cat]-stimulated GTP release measurements of selected mutants.

**Figure supplement 2.** Mutational tolerance of K-Ras[2-173] with a Q61L background mutation.

**Figure supplement 3.** Location of top-activating sites in the mammalian Ba/F3 cell experiment and structure of H-Ras, Rap1B, and Rab29.

**Figure supplement 4.** Sites of H-Ras where mutations have a different phenotype in the mammalian Ba/F3 cell and H-Ras+GAP experiments.

unregulated K-Ras, using the longer (K-Ras[2-173]) rather than the shorter (K-Ras[2-165]) construct (**Figure 3A** and **Figure 7—figure supplement 2B, C**). This result is in agreement with the stability measurement of Q61L in the H-Ras[1-166] construct (**Figure 4B**). Furthermore, the catalytic activity impairment produced by Gln 61 (**Hunter et al., 2015**; **Lu et al., 2016**) renders the GAP ineffective and the mutational sensitivity of the Ras+GAP condition looks identical to the unregulated condition. In conclusion, mutations at the cancer hotspots increase Ras signaling by breaking a regulatory mechanism rather than through destabilization.

## Gly 12, Gly 13, and Gln 61 are less conserved in the Ras superfamily than residues at stability-dependent gain-of-function sites

The nine most-activating sites of mutations in H-Ras[1-188] in the mammalian Ba/F3 cell experiment (**Figure 2A**) are Gly 12, Gly 13, Gln 61, and six stability-dependent gain-of-function sites (Val 14, Lys 117, Asp 119, Ser 145, Ala 146, and Phe 156). The locations of these nine sites in the Ras structure are shown in **Figure 7—figure supplement 3A**. We calculated the sequence diversity, measured as the Shannon entropy, of these nine sites in two different protein alignments: (i) the ~150 proteins that belong to the human Ras superfamily (**Rojas et al., 2012**), which includes a diverse set of proteins (e.g. Rap1B, Ral, Rab29) that share the same protein fold, and (ii) the proteins found in the alignment of metazoan Ras sequences used in our previous evolutionary study (**Bandaru et al., 2017**).

These nine sites are highly conserved across Ras orthologs in the metazoan Ras protein alignment (**Figure 7E**). However, in the Ras superfamily alignment, the six stability-dependent gain-of-function sites are more conserved than the cancer hotspots. Lys 117 and Asp 119 present the highest degree of sequence conservation, which we attribute to their critical role in nucleotide binding and stabilization. In contrast, Gly 12, Gly 13, and Gln 61 are replaced by other residues in many proteins of the Ras superfamily, such as Rab29, which has Ala instead of Gly at positions corresponding to residues 12 and 13 of human Ras, and Rap1B, which has Thr instead of Gln at position 61. The G-domains of Rab29 and Rap1B have sequence identities with H-Ras of ~27% and ~ 56%, respectively, and they maintain a similar three-dimensional structure (**Figure 7—figure supplement 3B**).

Our results are in agreement with the sequence conservation analysis conducted by Rojas and coworkers (**Rojas et al., 2012**), where they report that the sequence diversity of Gly 12, Gly 13, and Gln 61 within the Ras subfamily is high compared to other sites of biological relevance for Ras. The G12V, G13D, and Q61L variants of Ras are more stable than the K117N and D119A variants, which is consistent with the lower sequence conservation at the three cancer hotspots compared to the stability-dependent gain-of-function sites (**Figure 4C** and **Figure 7—figure supplement 2D**). Overall, these findings indicate that Gly 12, Gly 13, and Gln 61 residues are not structurally essential (**Buhrman et al., 2010**; **Franken et al., 1993**; **Khrenova et al., 2014**). Thus, given that GAP surveillance is the dominant layer of regulation in mammalian cells, the ability of mutations at these three residues to block the action of the GAP without seriously compromising the integrity of the protein fold makes them the perfect cancer hotspots.

## Concluding remarks

In our previous Ras study, we developed a bacterial two-hybrid assay to probe the mutational landscape of Ras in the presence and absence of its regulators (*Bandaru et al., 2017*). Here, we analyze the mutational profile of H-Ras in mammalian Ba/F3 cells and show that it resembles the mutational profile of Ras in the bacterial assay when a GAP is co-expressed. These results demonstrate that Ras is under constitutive surveillance by a GAP in Ba/F3 cells, although the identity of the specific GAP is unknown. Importantly, these experiments show that the bacterial experiments provide a generally reliable platform for the study of the mutational landscape of Ras, one that allows more rapid testing of different constructs and regulators than is practical with the mammalian system. We exploited the ease of experimentation in the bacterial system to reveal a link between the stability of Ras and the fitness profile of different mutations.

The variants that escape GAP surveillance and undergo activation in deep mutagenesis assays correlate well with variants found in cancer databases. However, the mammalian and bacterial mutagenesis data do not explain the predominance of mutations at Gly 12, Gly 13, and Gln 61 – the three cancer hotspots – in cancer databases. We determined that truncation of the C-terminal helix of Ras destabilizes H-Ras by ~7 kJ.mol$^{-1}$, and this destabilization alters the mutational sensitivity of the protein. Comparison of the mutational profiles of Ras constructs of different stabilities identifies a distinguishing feature between the three cancer hotspots and other sites where activating mutations are found in either the mutational screens or the cancer databases. The phenotype of mutations at the cancer hotspots remains virtually unaltered in different constructs, while other mutations that are activating in the context of the longer construct have a reduced level of activation or even a loss-of-function phenotype in the context of the shorter construct.

We purified and characterized three cancer hotspots variants – G12V, G13D, and Q61L – and two Ras variants found infrequently in cancer – K117N and D119A. We found that the G12V and Q61L mutations are not destabilizing, and G13D destabilizes H-Ras by just ~10 kJ.mol$^{-1}$, whereas the K117N and D119A mutations destabilize H-Ras by ~25 kJ.mol$^{-1}$. A mutation that decreases the stability of Ras can lead to activation by weakening nucleotide binding, thereby permitting spontaneous nucleotide exchange to reactivate Ras in the presence of the GAP. A reduction in nucleotide affinity further destabilizes Ras, since Ras is dependent on ligand coordination to maintain stability (*Zhang and Matthews, 1998a*; *Zhang and Matthews, 1998b*). When the shorter, less stable construct is used, the decrease in stability presumably leads to reduced levels of folded Ras. The same mutation may activate the longer Ras construct due to a higher stability threshold – the compensating stability being provided by the extension of the C-terminal helix of Ras (*Bershtein et al., 2006*).

Our findings regarding the coupling between C-terminal helix truncation and Ras stability allows a conceptual connection to be made with the mechanism of activation of heterotrimeric G proteins via G-protein-coupled receptors (GPCRs). Ras and the heterotrimeric G proteins share a similar G-domain architecture (*Flock et al., 2015*), act as molecular switches, bind to the membrane via lipid anchors (N-terminal in G-α and C-terminal in H-Ras), and bind the nucleotide through conserved motifs ($^{15}$GKS/T$^{17}$ and $^{116}$NKXDL$^{120}$ in H-Ras). The activation of heterotrimeric G proteins by GPCRs involves allosteric release of GDP by modulation of the C-terminal helix of the G domain (*Bos et al., 2007*; *Flock et al., 2015*). Given the structural and functional similarities between the two proteins, we suspect that nucleotide binding in Ras is also coupled to its C-terminal helix. The changes in stability and mutational sensitivity of Ras constructs in which the length of the C-terminal helix is varied are a manifestation of such coupling.

Conservation analysis of the Ras superfamily revealed that sites where mutations are activating in the Ba/F3 assay but infrequent in cancer (e.g. Val 14, Lys 117, and Asp 119) have higher sequence conservation than cancer hotspots. The conservation of the residues at the sites of stability-dependent activating mutations across the Ras superfamily indicates that the wild-type versions of these residues play essential roles in maintaining the Ras fold and coordinating the nucleotide. In contrast, Gly 12, Gly 13, and Gln 61 play roles specialized for interaction with Ras-GAP and catalyzing nucleotide hydrolysis, enabling mutations at these three sites to activate Ras without severe consequences for the stability of the Ras fold or nucleotide coordination.

Another connection between Ras activation and stability emerges from our analysis of hydrogen-deuterium exchange rates by NMR for three activating H-Ras variants (H27G, L120A, and Y157Q). These variants display increased nucleotide exchange rates (*Bandaru et al., 2017*) and exhibit

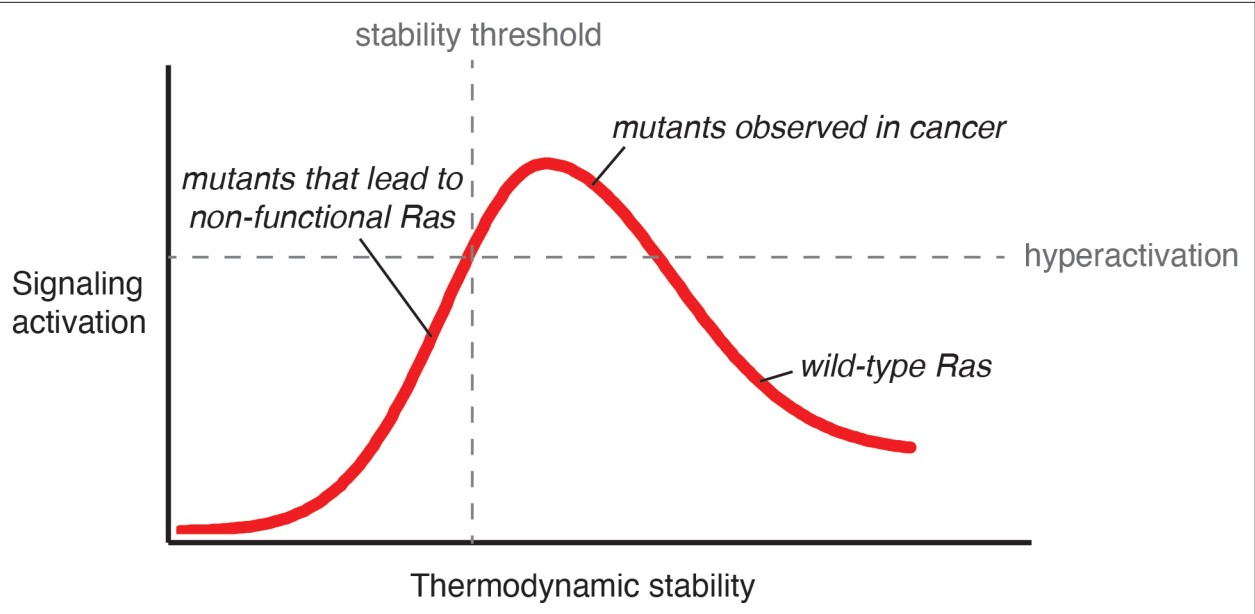

**Figure 8.** Signaling activity versus thermodynamic stability. A conceptual relationship between the thermodynamic stability and signaling activity of Ras variants is shown. A decrease in stability can increase signaling activity by enhancing nucleotide-exchange rates or compromising catalytic activity. However, further stability reduction reaches the stability threshold. The stability threshold buffers the deleterious effects of mutations (*Bershtein et al., 2006*), and is epistatic in nature; once exhausted, the deleterious effects of mutations become fully pronounced. Hence, signaling activity decreases. The closer a mutant is to the stability threshold, the narrower the window where the mutant can promote hyperactivation *in vivo*.

increased HDX, consistent with the destabilization of the structure. These three residues appear to occupy important positions that 'lock down' the structure and enhance coupling across the protein (*Bandaru et al., 2017*). We surmise that the release of this coupling affects the nucleotide-binding site, resulting in increased HDX.

Activating mutations that decrease conformational stability and accelerate GDP dissociation have been reported for Rac1a and G-α GTPases (*Toyama et al., 2019a*; *Toyama et al., 2019b*; *Toyama et al., 2017*). The high affinity of GTPases for GDP prevents GEF-independent activation in the absence of appropriate cellular stimuli. However, destabilizing mutations that loosen the grip on the nucleotide increase nucleotide turnover, and our results indicate that Ras has a substantial innate capacity to be activated by such mutations. The overall stability of the Ras fold determines the balance between increases in signal output and decreases in signaling capacity due to the unfolding of the protein (*Figure 8*).

This interplay between stability and activity is also observed in other types of signaling proteins. For example, protein kinases are activated by mutations that destabilize them and thereby break regulatory mechanisms, as seen for the V600E cancer mutation in B-Raf (*White et al., 2018*). The action of the chaperone HSP90 protects the destabilized kinases from unfolding and subsequent proteasomal degradation, allowing the mutant kinases to promote hyperactivation (*Taipale et al., 2012*). Ras proteins do not appear to be protected by such chaperones, allowing differences in non-specific chaperone activity between cell types to determine the extent to which destabilizing mutations promote activation. The stability of Ras can also be impacted by factors such as post-translation modifications and membrane-induced mechanical stress (*Campbell and Philips, 2021*; *Gavrilov et al., 2015*; *Kapoor et al., 2013*; *Zhang and Matthews, 1998a*; *Zhang and Matthews, 1998b*). A combination of these effects might explain why destabilizing mutations are less frequent in cancer and why there are tissue-specific patterns of activating mutations in Ras (*Cook et al., 2021*; *Li et al., 2018*).

## Materials and methods

### Construction of the Ras variant DNA libraries

Ras libraries for four constructs (H-Ras with residues 2–166, H-Ras with residues 2–180, K-Ras with residues 2–165, and K-Ras with residues 2–173) containing every possible amino acid substitution mutant were generated by using oligonucleotide-directed mutagenesis as described in *Bandaru et al., 2017*. First, two mutagenic primers with 15 base pair sequence complementarity on each side of the targeted codon were generated. The primers contain random codons for each of the residues in the respective constructs, allowing all 20 amino acids to be represented in each position (*Bandaru et al., 2017*). Library generation required two sets of polymerase chain reactions (PCRs) to produce the full-length double-stranded product containing a single degenerate codon (*Bandaru et al., 2017*). Each PCR product was then run on an agarose gel and purified using a gel extraction protocol (NEB). The purified PCR product concentrations were measured using the Picogreen assay (Thermo Fisher), pooled in equimolar concentrations, digested with HindIII (NEB) and Bsu36I (NEB), and ligated with T4 DNA ligase (Invitrogen) into RNA-α vector. The ligated product was then transformed into NEB 10-beta *E. coli* cells through heat shock transformation. The NEB 10-beta cells were then grown on Kan + LB agarose plates to assess the efficiency of the transformation. The rest of the cells were grown in a 5 ml overnight culture in LB and Kanamycin. The DNA from these cells was extracted using Qiagen miniprep kit and sequenced with Illumina sequencing to ensure that every Ras variant was equally represented. We limited the Illumina sequencing to 500 base paired-end reads to ensure high-quality base calls. Thus, each Ras library was generated as two or three separate sub-libraries. Variants for H-Ras$^{2-166}$ were divided into residues 2–56, 57–111, and 112–166. Variants for K-Ras$^{2-165}$, K-Ras$^{2-173}$, and H-Ras$^{2-180}$ were divided into residues 2–88 and 89-end.

### Ba/F3 cell culture and retroviral transduction

Ba/F3 cells (a generous gift from Neil Shah, UCSF) were maintained in complete RPMI 1640 medium (Gibco) with glutamine supplemented with 10% fetal bovine serum (VWR), 100 units/mL penicillin-streptomycin (Gibco), and 2 ng/mL recombinant murine interleukin-3 (IL-3) (Gibco). Cells were grown in a humidified incubator at 37 °C and 6% atmospheric $CO_2$. Cells were examined for IL-3 dependence regularly, ensuring that cells cultured without IL-3 died within 24 hr. Mutant H-Ras libraries used for the bacterial-two-hybrid assay (*Bandaru et al., 2017*) were subcloned into pMIG, a murine stem cell virus retroviral transfer vector harboring an internal ribosome entry site, and a green fluorescent protein cassette (IRES-GFP). The resulting libraries were transfected with pCL-Eco (Addgene #12371) into HEK 293T cells with Lipofectamine 2000 (Gibco). Media was harvested after 48 hr, applied to a 0.45 μM syringe-driven filter, and used to transduce Ba/F3 cells at a multiplicity of infection (MOI) of 0.1. After confirming GFP expression by flow cytometry, cells were grown for 1 day in the presence of IL-3. Then, a fraction of the cells were used as the 'nselected' population. The remainder of the cells ('selected' population) was washed thoroughly with PBS and grown for 7 days in media without IL-3. Both unselected and selected cells were harvested by centrifugation and resuspended into 200 μL of PBS and processed with a DNeasy Blood and Tissue kit (Qiagen) per the manufacturer's instructions to obtain genomic DNA. The extracted DNA was used to generate libraries for Illumina sequencing as described below.

### Bacterial two-hybrid selection assay

We adopted the protocol described in *Bandaru et al., 2017*. Electrocompetent MC4100-Z1 cells containing both the pZS22 and pZERM1-CAT plasmids were transformed with 100 ng of the pZA31 plasmids containing the libraries for H-Ras (construct length 2–166), H-Ras (construct length 2–180), K-Ras (construct length 2–165), and K-Ras (construct length 2–173), cultured for one hour in LB media. Small aliquots were plated LB-agarose plates with kanamycin, ampicillin, and trimethoprim to test for transformation validity. The remainder of each culture was grown overnight in LB media containing 20 μg/mL trimethoprim, 50 μg/mL kanamycin, and 100 μg/mL ampicillin. The following morning, 30 μL of each culture was diluted to an optical density (OD) at 600 nm of 0.001. The diluted cultures were grown for 2 hr in 10 mL of LB at the same antibiotic concentrations. Next, cells were diluted to an OD of 0.0001 and induced with 50 ng/μL doxycycline and 100 μM IPTG for 3 hr in 60 mL of LB + antibiotics. After induction, 50 mL of culture was reserved for the Illumina sequencing of the pre-selection population (*Bandaru et al., 2017*). Selection cultures were started with the remainder of the induction

cultures at an OD of 0.001 in 60 mL of LB + antibiotics + inducer with 75 µg/mL chloramphenicol for 5 hr. Then, the second set of selection cultures was started by diluting the previous selection cultures to an OD of 0.001 in a new batch of 60 mL of LB + antibiotics + inducer with 75 µg/mL chloramphenicol for 4 hr. Two selection steps were used to ensure that the OD remained below 0.1 throughout the experiment. Both pre- and post-selection cultures were spun down and stored frozen at –80 °C. The bacterial datasets were obtained from new experiments conducted for this work, except for H-Ras[2-166]+GAP, where the raw data from our earlier work was reprocessed (*Bandaru et al., 2017*).

## Next-generation sequencing using the Illumina MiSeq platform

Next-generation sequencing was used to determine the frequencies of each variant in the experiments. For using the MiSeq instrument (Illumina), we followed the recommended protocols described in the MiSeq system guide (Document # 1000000061014 v00, July 2018, and Document # 15039740 v10, February 2019). Pre- and post-selection cultures of each library were then miniprepped using Qiagen's standard protocol. The extracted DNA was then amplified by PCR and attached to barcode oligonucleotides compatible with the MiSeq sequencing platform (Illumina, index1: i7 primers, and index: i5 primers). The barcoded samples were then pooled together at equimolar concentrations and diluted to 4 nM. 5 µL of the 4 nM pooled sample were then combined with 5 µL of 0.2 N NaOH, centrifuged, and combined with 990 µL of prechilled HT1 (hybridization buffer, Illumina) to create 1 ml of a 20 pM denatured DNA sample (Illumina). PhiX control library (Illumina) was added until the concentration was 20 pM. The sample was diluted to 12 pM by adding more HT1, then loaded onto a reagent cartridge (MiSeq Reagent kit v2 300-cycles or 500 cycles) and sequenced. Raw DNA reads were assembled using PEAR (Paired-End reAd mergeR) version 0.9.6 (*Zhang et al., 2014*), with a minimum quality threshold of 30. Then, the barcode adapters were trimmed using cutadapt version 1.7.1 (*Martin, 2011*).

## Calculation of enrichment scores using mutagenesis-visualization software

Data processing, statistical analysis, and visualization were conducted using 'mutagenesis-visualization' version 1.0, an open-source Python package that can analyze saturation-mutagenesis datasets (*Hidalgo et al., 2021a*). The documentation can be found at readthedocs, the source code on GitHub (*Hidalgo et al., 2021b*), and the software can be used without previous Python knowledge. The graphs in this work have been generated using mutagenesis-visualization. The raw data generated in this work can be found on GitHub (copy archived at swh:1:rev:66bd65d37428007210fe07da83369a566c6cb18b, *Hidalgo, 2022*).

First, the trimmed DNA reads of each pre- and post-selection cultures of each library for H-Ras (construct length 2–166), H-Ras (construct length 2–180), K-Ras (construct length 2–165), and K-Ras (construct length 2–173) were counted. Then, a baseline count correction step was conducted on the post-selection samples (except in Ras+GAP experiments, see the section below). Next, the relative enrichment ($E_x^i$) was calculated using *Equation 1*.

The first term of *Equation 1* is the logarithm of the ratio of counts (c) of observing codons representing each amino acid $x$ at each position $i$ in the selected and unselected samples. The second term of *Equation 1* is the median of the ordered list of logarithms of the elements of the vector obtained by conducting pair-wise division, denoted ⊘, between the selected and unselected counts ($C^{wt,selected}$ and $C^{wt,unselected}$, respectively) for the variants that are synonymous with the wild-type (wt) allele. A Ras variant with an enrichment score of zero propagates in the assay at the same rate as the wild-type variants. Variants with scores of ±1 propagate tenfold faster or slower than wild-type variants, respectively. This form of calculating the enrichment scores is an improvement over the traditional calculation, where only the counts for the wild-type reference DNA sequence were used to center the data (*Equation 3a and 3b*).

$$\Delta E_x^i = log_{10}\left[\frac{c_i^{x,sel}}{c_i^{x,unsel}}\right] - log_{10}\left[\frac{c_i^{wtreference,sel}}{c_i^{wtreference,unsel}}\right] \tag{3a}$$

In saturation-mutagenesis experiments, there are different sources of error, such as from the next-generation sequencing procedure, PCR amplification steps, counting statistics, or experimental conditions during the selection process (e.g. changes in antibiotic concentration from replicate to replicate).

Hence, relying on a single data point to center the data makes enrichment scores calculated using *Equation 3a and 3b* more sensitive to outliers. To overcome this limitation, we use the synonymous wild-type alleles to obtain more data points (*Equation 1*). These alleles comprise all the sequences in the DNA library that differ in DNA sequence to the wild-type gene used as the template to create the library, yet they translate to the same protein. In the Ras libraries used in this paper, there are more than a hundred synonymous wild-type alleles. Scatter plots of the wild-type allele enrichment scores for the pair-wise combinations of replicates show low $R^2$ values (see documentation). The absence of correlation between the enrichment values of wild-type alleles in replicates rules out the possibility that any particular wild-type DNA sequences is more enriched than another wild-type sequence due to expression level differences. This topic is covered in the documentation of mutagenesis-visualization (*Hidalgo et al., 2021a*).

Afterward, synonymous variants were combined using the mean enrichment score. The result is a unimodal Gaussian distribution of enrichment scores with a left/right shoulder (depending on the sample). The data were centered at zero using the mode of the enrichment scores of each DNA sequence that translates to wild-type Ras. The last step in the data processing was a scaling step, where we set the standard deviation of each distribution to be constant, thus allowing for comparison between different datasets.

## Baseline correction step in enrichment score calculation

We introduce a correction to account for cells that do not express functional Ras molecules but grow at some non-zero rate in the bacterial assay. That is, there is some leaky expression of the antibiotic resistance gene. In order to obtain growth rates for variants between the limits of 0 the growth rate of the organism without antibiotic exposure, we perform a baseline correction.

Mutagenesis-visualization performs a baseline correction by using the counts of the stop codons present in the selected library. First, the counts of each amino acid $x$ at each position $i$ in the unselected sample ($c_i^{x,unsel}$), multiplied by the median of the stop codon frequencies ($\tilde{x}_{f^*}$) are subtracted from the counts in the selected sample ($c_i^{x,sel}$) (*Equation 3a and 3b*). Then, the 'corrected' selected counts ($c_{I,corrected}^{x,sel}$) are used to calculate the enrichment scores ($E_x^i$) using *Equation 1*.

$$c_{i,corrected}^{x,sel} = c_i^{x,sel} - c_i^{x,unsel}\widetilde{\tilde{x}_f}; \ \tilde{x}_f = median\ of\left(\frac{C^{sel}}{C^{unsel}}\right) \tag{3b}$$

While the correction does not affect the relative order of the variants with respect to the population, it changes the shape of the enrichment score distribution (see documentation of mutagenesis-visualization). Before the baseline correction, the enrichment scores distribution is bimodal, and the new distribution is unimodal. This baseline correction cannot be applied if the growth rate of the stop codon variants is the same as most of the other variants (i.e. Ras+GAP datasets).

## Calculation of receiver operator characteristic (ROC) curves

Receiver operating characteristic (ROC) curves were used to make quantitative comparisons between the saturation-mutagenesis datasets and the COSMIC (https://cancer.sanger.ac.uk/cosmic) and gnomAD (https://gnomad.broadinstitute.org/) databases or between two saturation-mutagenesis datasets. When using the COSMIC database v94 as the reference, each variant with five or more counts in at least one isoform dataset was considered a true positive. Because we set a minimum of five counts to consider a mutation in the COSMIC database, we eliminate low-frequency mutations present in the COSMIC database that may not be pathogenic. The variants found in the gnomAD were considered true negatives (*Livesey and Marsh, 2020*). For gnomAD, we used the H-Ras, N-Ras, and K-Ras non-cancer datasets versions 2.1.1 and 3.1.2 and filtered out any mutation labeled as 'pathogenic'. When a variant is found in gnomAD and COSMIC datasets, it is considered a true positive. Some pathogenic mutations may be present in the individuals screened for the gnomAD database, and the pathogenic label has not been added.

We had a total of 59 true positives and 152 true negatives. We do not use mutations absent in the COSMIC database as true negatives to prevent misclassifying mutations that can activate Ras and activate cell senescence (*Li et al., 2018*). Furthermore, some mutations may not yet be seen in the COSMIC database because they require more than one base-pair change. We performed the ROC analysis in two ways: using K-Ras counts, and combining the counts for the three isoforms, so we had

more data points. The results were unaffected. We include the list of variants used for this analysis in the supplementary data and the scripts used.

We repeated the ROC analysis using the oncogenic mutations found in the cBioPortal (http://www.cbioportal.org/) as the true positives (*Cerami et al., 2012*; *Gao et al., 2013*). This portal hosts many cancer studies, including the AACR GENIE Project Data version 10.1, with 123736 samples (*AACR Project GENIE Consortium, 2017*). It also hosts several other studies, from where we obtained more Ras variant frequencies in cancer. We entered the cBioPortal 'data sets' tab to obtain the data and selected the 'curated set of non-redundant studies'. This curated set contains 178 studies and 57,523 samples. We submitted a query with the genes 'KRAS, HRAS, NRAS' and downloaded the list of mutations for all the studies combined. Then, we used the five counts cutoff to consider mutations for the true positive list. The rest of the ROC analysis was done as described above, replacing the list of true positives from COSMIC with the new list derived from cBioPortal. The gnomAD mutations list was used as the true negatives.

When using a second saturation-mutagenesis dataset as the reference, mutations with a score of >1.5 times the standard deviation are considered true positives, and the rest as true negatives. The ROC curve is generated by graphing the fraction of true positives versus the fraction of false positives at various threshold settings, and the estimated area under the curve (AUC) gives a measure of the overall accuracy of the predictions.

The COSMIC database includes a FATHMM (Functional Analysis through Hidden Markov Models) pathogenicity score for each variant. We have scored the variants present in the COSMIC database using our best-predicting dataset, that is, H-Ras[1-188] in Ba/F3 cells, so the scores can become a new metric used to score the pathogenicity of oncogenic mutations analogous to FATHMM (fhidalgor/ras_cancer_hidalgoetal).

## Protein expression and purification for stability measurements

Wild-type H-Ras of construct lengths 1–166 and 1–173, wild-type K-Ras of construct length 1–173, and H-Ras variants K117N and D119A (both of construct lengths 1–173), were all tagged at the N-terminus with a hexahistidine tag, and transformed into 100 µL *Escherichia coli* (BL21 (DE3)) in the pProEX expression vector. A starter culture of 50 mL Terrific Broth (TB) with 100 µg/mL ampicillin was grown, with 220 rpm shaking, at 37 °C overnight. After bacterial growth to an OD600 of 0.5 in TB containing ampicillin at 37 °C, induction was carried out at 18 °C with 0.4 mM IPTG. The bacteria were further allowed to grow, shaking at 220 rpm, at 18 °C for 18 hr. The bacteria were pelleted by centrifugation, and the pellets were resuspended in Ni-A buffer (20 mM tris, 1 M NaCl, 1 M urea, 3 mM MgCl$_2$, 0.5 mM TCEP, pH 8, filter sterilized) on ice and either stored at –80 °C or used immediately for subsequent purification steps.

A protease inhibitor cocktail (Roche) was added to resuspended protein pellets. The bacteria were lysed through sonication, and cell debris was removed by centrifugation. The supernatant was applied to a 5 mL nickel column (HisTrap FF columns, GE Healthcare). The column was washed with 15 mL of W1 buffer (Ni-A + 20 mM imidazole), 15 mL of W2 buffer (Ni-A + 40 mM imidazole), and 15 mL W3 buffer (Ni-A + 60 mM imidazole). The protein was then eluted from the nickel column with elution buffer (Ni-A + 400 mM imidazole). The protein was concentrated to ~1.5 mL in Ni-A buffer, then diluted into Ni-A to remove imidazole. The hexahistidine tag was then cleaved overnight using hexahistidine-tagged TEV protease (1 mg TEV per 25 mg crude Ras) to leave a single N-terminal glycine scar. After cleavage for 48 hr, the protein solution was run over a 5 mL nickel column equilibrated with 20 mL Ni-A buffer, and the flow-through was collected.

For CD spectroscopy measurements, the flow-through was concentrated to 1 mL and further purified using a pre-equilibrated Superdex 75 10/300 GL column in 20 mM HEPES, 150 mM NaCl, 5 mM MgCl2, 0.5 mM TCEP, pH 7.2. Collected fractions were concentrated, and concentration was estimated using absorbance at 280 nm.

For pulse proteolysis experiments, the flow-through was concentrated to 1.5 mL in assay buffer with GDP (20 mM tris, 5 mM MgCl2, 140 mM NaCl, 0.5 mM TCEP, 100 µM GDP, pH 7.2, filter sterilized). 10 µL of 12 x and 120 x diluted protein prep was then loaded into a 15% SDS-PAGE gel (BioRad) with BioRad Precision Plus unstained standard for comparison. After running at 200 V for 30 min, the gel was stained with SYPRO Red (50 mL 7.5% v/v acetic acid, 5000 x SYPRO Red stock) and shaken overnight. Finally, the gel was imaged with the UV transilluminator to quantify the total protein concentration.

## CD measurements

Fresh assay buffer with 10 M urea was used to set up 8 M and 0 M urea stocks with 35.5 µM GDP and 0.5 µM Ras. 2.7 mL samples with urea concentrations ranging from 0 to 8 M were set up by mixing these stocks, then incubated overnight at 25 °C. Spectra of the 0 M and 8 M samples showed the greatest ratio of signal-to-noise change at 228 nm, so this wavelength was used for all urea titrations, with a bandwidth of 2.5 nm. Each sample was stirred as four 15-s measurements were taken on an Aviv 410 CD spectrometer in a 1 cm cuvette at 25 °C. The refractive index of each sample was measured with a refractometer (Zeiss), and the difference in refractive index from the no-urea sample was used to estimate the urea concentration (*Pace, 1986*). Each titration yielded a maximum likelihood fit to a model with linear unfolded and folded baselines and normal noise, where the scale parameter is a linear function of urea concentration (*Equation 4*; *Carpenter et al., 2017*):

$$CD\ signal_i \sim Normal\left(\left(signal_{unfolded} + slope_{unfolded}\cdot[urea]_i\right) + \left(signal_{folded} + slope_{folded}\right.\right.$$
$$\left.\left.\cdot[urea]_i\cdot\left(\frac{e^{\frac{-m\cdot[urea]_i+\Delta G}{RT}}}{1+e^{\frac{-m\cdot[urea]_i+\Delta G}{RT}}}\right)\right)\sigma_{constant} + \sigma_{proportional}\cdot[urea]_i\right) \tag{4}$$

## Proteolysis measurements

In order to be able to measure the unfolding of unstable Ras mutants, we increased the GDP concentration from 36 to 100 µM, adding ~2.53 kJ.mol⁻¹ to the value of $\Delta G_{unf}$. For this reason, the reported $\Delta G_{unf}$ values for H-Ras¹⁻¹⁶⁶ and H-Ras¹⁻¹⁷³ are slightly higher in the proteolysis measurements than in the CD measurements (*Figure 4A*). The change in ligand contribution to the stability is shown in *Equation 5*.

$$Ligand\ contribution = RT\left(log_{10}\left[\frac{1+\frac{[GDP]_{new}}{K_D}}{1+\frac{[GDP]_{old}}{K_D}}\right]\right) \tag{5}$$

In *Equation 5*, $K_D$ represents the dissociation constant of GDP, and [GDP]$_{new}$ and [GDP]$_{old}$ represent the GDP concentrations used in the proteolysis and CD measurements, respectively. Ligand contribution remains roughly independent of decreased mutant affinity even up to a $K_D$ of 3 µM, where there is a 5%, or 0.13 kJ.mol⁻¹ deviation from this ideal behavior. Hence, we can continue to compare the stability of mutants even with the drastically reduced nucleotide affinities we expect of some point mutants.

Fresh assay buffer with 10 M urea was used to prepare 8 M and 0 M urea stocks with 100 µM GDP and 1.6 µM Ras. In a 96-well plate, twelve 100 µL urea concentrations were created by mixing these stocks, increasing from 0 M urea to 8 M urea. The well plate was sealed with parafilm and incubated overnight at 25 °C. Next, 1 µL of 1 M CaCl₂ was added to each well, followed immediately by 2 µL thermolysin stock (10 mg/mL lyophilized thermolysin by absorbance at 280 nm, Sigma Aldrich, in 2.5 M NaCl, 10 mM CaCl₂, 0.2 µm, filtered) and incubated for 60 s (*Park and Marqusee, 2006*). The reaction was quenched by addition to 50 mM EDTA (pH 8, filter sterilized). Next, 6 x SDS-PAGE loading buffer was added to each quenched solution, and the proteolysis samples were denatured in the 96-well thermocycler at 98 °C for 5 min. Next, 10 µL of each quenched solution was loaded into 15%, 15-well 4–20 x SDS-PAGE gels (BioRad), and the system was run on ice at 60 V for 35 minutes, then at 100 V for 1.5 hr. Gels were stained with 5000 x SYPRO Red for 1 hr, incubated in H₂O overnight to eliminate background signal, and then imaged with a UV transilluminator. Urea concentrations were estimated from each quenched sample as for CD. The proteolysis sample band intensities were quantified with ImageJ. Each titration yielded a maximum likelihood fit to a model with constant unfolded and folded baselines, an m-value constrained to its CD-estimated value, and normal noise with constant scale parameter (equivalent to nonlinear least squares), that is *Equation 6*:

$$Band\ signal_i \sim Normal\left(signal_{unfolded} + signal_{folded}\cdot\left(\frac{e^{\frac{-m_{CD}\cdot[urea]_i+\Delta G}{RT}}}{1+e^{\frac{-m_{CD}\cdot[urea]_i+\Delta G}{RT}}}\right), \sigma\right) \tag{6}$$

### *In vitro* nucleotide exchange assay

GDP release assays were performed with Ras bound to mant-GDP (Axxora Biosciences), loaded with the protocol described in our previous Ras study, and the purified H-Ras mutants and SOS[cat] were used from that previous study (*Bandaru et al., 2017*). Each H-Ras mutant was loaded with mant-GDP by incubating the protein with a ten-fold molar excess of EDTA in the presence of a ten-fold molar excess of GTP. Next, the loading reaction was performed on ice for one hour, and the reaction was quenched by adding a 20-fold molar excess of $MgCl_2$. Then, the reaction was buffer exchanged using a NAP-5 column (GE Healthcare) to remove excess nucleotide and salt. Then, 1.5 µM H-Ras was mixed with 2.5 µM SOS[cat] and 3.5 mM GDP in solution in the following buffer: 150 mM NaCl, 40 mM HEPES, 4 mM $MgCl_2$, 1 mM TCEP, and 5% glycerol. Nucleotide exchange rates were monitored on a Tecan fluorescence plate reader, taking measurements every 15 s. Data were fit to a single exponential decay curve for quantification of GDP exchange rates.

### Protein preparation for hydrogen-deuterium exchange experiments

Wild-type H-Ras and four mutants (H27G, Q99A, L120A, and Y157Q), all with construct length 1–166, were expressed and purified as follows. Each construct was transformed into BL21(DE3) *E. coli* and plated on kanamycin-containing plates. These cultures were then used to inoculate a liquid culture containing 100 µg/mL kanamycin in either Terrific Broth (TB) or minimal media containing either an $N^{15}$ source or $N^{15}$ and $C^{13}$ sources, depending on the experiment. These cultures were grown at 37 °C to an optical density of 0.8–1 and then were induced with 1 mM Isopropyl β-D-thiogalactopyranoside (IPTG). Growth continued overnight at 18 °C. The next day, the bacteria were pelleted and resuspended in a small volume of Ni-NTA buffer (500 mM NaCl, 20 mM Tris-HCl pH 8.5, 20 mM Imidazole, 5% Glycerol) and flash frozen.

To begin the protein preparation, cell pellets were thawed and lysed by sonication or homogenization at 4 °C, depending on the cell pellet volume. This lysate was then spun for an hour at 16,500 rpm to separate soluble and insoluble fractions. The soluble fraction was collected and flowed through a HisTrap FF column (GE Healthcare). The column was washed with 10 column volumes (CV) Ni-NTA, followed by 10 CV low-salt Ni-NTA (50 mM NaCl, 20 mM Tris-HCl pH 8.5, 20 mM Imidazole, 5% Glycerol). This column was then attached directly to a HiTrap Q FF column (GE Healthcare) equilibrated in low-salt Ni-NTA elute buffer (50 mM NaCl, 20 mM Tris-HCl pH 8.5, 500 mM Imidazole, 5% Glycerol). Protein was eluted onto the Q column with 10 CV low-salt Ni-NTA elute. The Q column was then washed with a salt gradient in Q buffer (50 mM – 1 M NaCl, 20 mM Tris-HCl pH 8.5, 5% Glycerol) over 10 CV, and then 5 CV of the high-salt Q buffer. The protein eluted in a single peak during gradient. This sample was collected and incubated at 4 °C overnight with TEV protease to remove the TEV-linked His-tag.

After overnight incubation, the sample was run through a HisTrap column, this time only using high-salt Ni-NTA buffer, and flow through and wash were collected. This sample was concentrated to less than 1 mL. The sample was incubated at room temperature for 2 hr with 3 x molar excess GMP-PNP, 1 mM $ZnCl_2$, and 10 U bovine alkaline phosphatase. This sample was further purified by gel filtration over a Superdex 75 column (10/300 GL, GE Healthcare) in gel filtration buffer (40 mM HEPES, 150 mM NaCl, 4 mM $MgCl_2$, 1 mM TCEP, and 5% glycerol at pH 7.4).

### Hydrogen-deuterium exchange experiments

$N^{15}$ labeled protein was prepared at 500 µM in gel filtration buffer and divided into 500 µL aliquots. These aliquots were flash-frozen, lyophilized, and then stored at –80 °C. Just before beginning the experiment, the sample was dissolved in 500 µL $D_2O$. The sample was then loaded into an 800 MHz NMR at 25 °C, and the first HSQC was taken immediately. This HSQC was repeated six times, with no delay between spectra, followed by six more spectra with a delay of 3 hr between the start of each experiment.

Each spectrum was baseline corrected and phased. Peaks were assigned based on BMRB 17678. Assignments were validated by HNCA carried out with $C^{13}/N^{15}$ labeled WT protein. Each peak was fit and integrated. The volume decay was fit to a single exponential, if possible. Fits with $R^2 > 0.7$ were kept for analysis.

We also carried out HDX by mass spectrometry. Deuterated buffer was prepared by lyophilizing Ras gel filtration buffer (40 mM Hepes pH 7.4, 150 mM NaCl, 4 mM MgCl2, 5% Glycerol) overnight

and resuspending in equivolume D$_2$O (Sigma-Aldrich 151882). This process was repeated twice before storing the lyophilized buffer at –80°C. Lyophilized buffer was resuspended in equivolume D$_2$O before use.

Aliquots of WT Ras were thawed on ice and diluted to a final concentration of 10 μM. To initiate hydrogen exchange, 10 μM WT Ras was diluted into deuterated buffer to a final concentration of 1 μM. Exchanging samples were kept in a temperature-controlled block maintained at 25°C. At various time points, 30 μL of the exchanging reaction were added in a 1:1 ratio to quench buffer (1.5 M glycine, 3.5 M GdmCl, pH 2.5) on ice before flash freezing samples in liquid N$_2$. Samples were stored at –80°C for subsequent LC/MS analysis.

LC/MS was performed as described in previous work (*Costello et al., 2021*). Briefly, samples were thawed and then injected into a valve system (Trajan LEAP) coupled to an LC (Thermo Ultimate 3000). Sample time points were injected in a non-consecutive order. The valve chamber, trap column, and analytical column were kept at 2°C, and the protease chamber at 4°C. Upon injection, samples underwent proteolysis using two in-line columns manually packed with aspergillopepsin (Sigma-Aldrich P2143) or porcine pepsin (Sigma Aldrich P6887) with buffer A (1% formic acid) flowing at a rate of 200 μL/min. Peptides were desalted for 4 min on a trap column (1 mM ID x 2 cm, IDEX C-128) manually packed with POROS R2 reversed-phase resin (Thermo Scientific 1112906). Peptides were separated on a C8 analytical column (Thermo Scientific BioBasic-8 5 μm particle size 0.5 mM ID x 50 mM 72205–050565) with buffer B (90% acetonitrile, 0.1% formic acid) flowing at a rate of 40 μL/min and increasing from 10% to 45% for the first 14 min, and then from 45% to 100% over 30 s. Eluted peptides were then identified via MS, and analytical and trap columns were washed using a sawtooth gradient. Protease columns were washed twice with 100 μL of 1.6 M GdmCl, 0.1% formic acid prior to each subsequent injection.

Exchange for WT Ras was performed in duplicate to verify that the EX1 behavior in the N terminus was reproducible. Two MS/MS experiments were carried out on undeuterated samples to generate peptide lists with the MS settings described in *Costello et al., 2021*, using Byonic (Protein Metrics) software. Peptide isotope distributions were fit using HDExaminer 3. The centroids of the observed distributions for each time point and each peptide were then exported for plotting and figure generation in Jupyter python notebooks.

## Acknowledgements

We thank members of the Kuriyan lab for helpful discussions. We thank Subu Subramanian for helpful discussions regarding the data processing and analysis, and Kendra Marcus for insightful discussions about Ras biology. We also thank Joseph Paul and Tim Eisen for helping to review the manuscript. We thank Rama Ranganathan for helping us adapt the bacterial-two-hybrid system to study the Ras cycle. This work was supported in part by NIH grant P01AI091580-09 awarded to John Kuriyan. The authors would like to acknowledge the American Association for Cancer Research and its financial and material support in the development of the AACR Project GENIE registry, as well as members of the consortium for their commitment to data sharing. Interpretations are the responsibility of study authors.

## Additional information

### Competing interests

Deepti Karandur: Early Career Reviewer, eLife. The other authors declare that no competing interests exist.

### Funding

| Funder | Grant reference number | Author |
| --- | --- | --- |
| National Institutes of Health | P01AI091580-09 | John Kuriyan |

| Funder | Grant reference number | Author |
| --- | --- | --- |

The funders had no role in study design, data collection and interpretation, or the decision to submit the work for publication.

## Author contributions

Frank Hidalgo, Conceptualization, Data curation, Formal analysis, Investigation, Methodology, Software, Validation, Visualization, Writing – original draft, Writing – review and editing; Laura M Nocka, Data curation, Formal analysis, Investigation, Methodology, NMR mutants, NMR mutants, NMR mutants, Visualization; Neel H Shah, Conceptualization, Investigation, Methodology, Writing – review and editing; Kent Gorday, Data curation, Methodology, Software, Visualization; Naomi R Latorraca, Data curation, MS, MS, Methodology; Pradeep Bandaru, Methodology; Sage Templeton, Data curation, Formal analysis, Methodology; David Lee, Investigation, Methodology; Deepti Karandur, Conceptualization, Supervision, Writing – original draft; Jeffrey G Pelton, Methodology, NMR mutants, NMR mutants, NMR mutants, Resources; Susan Marqusee, MS, MS, Methodology, Supervision; David Wemmer, Conceptualization, Methodology, NMR mutants, NMR mutants, NMR mutants, Resources; John Kuriyan, Conceptualization, Funding acquisition, Investigation, Project administration, Supervision, Writing – original draft, Writing – review and editing

## Author ORCIDs

Frank Hidalgo (iD) http://orcid.org/0000-0003-1700-2697
Neel H Shah (iD) http://orcid.org/0000-0002-1186-0626
Pradeep Bandaru (iD) http://orcid.org/0000-0002-9354-3340
Deepti Karandur (iD) http://orcid.org/0000-0002-6949-6337
Susan Marqusee (iD) http://orcid.org/0000-0001-7648-2163
David Wemmer (iD) http://orcid.org/0000-0001-6252-3390
John Kuriyan (iD) http://orcid.org/0000-0002-4414-5477

## Decision letter and Author response

Decision letter https://doi.org/10.7554/eLife.76595.sa1
Author response https://doi.org/10.7554/eLife.76595.sa2

# Additional files

## Supplementary files

• Transparent reporting form

## Data availability

All data generated or analyzed during this study are included in a GitHub repository named "https://github.com/fhidalgor/ras_cancer_hidalgoetal", (copy archived at swh:1:rev:66bd65d37428007210fe07d-a83369a566c6cb18b). The dataset was also uploaded to zenodo (DOI: https://doi.org/10.5281/zenodo.6131510).

The following dataset was generated:

| Author(s) | Year | Dataset title | Dataset URL | Database and Identifier |
| --- | --- | --- | --- | --- |
| Hidalgo F, Nocka LM, Shah NH, Gorday K, Latorraca NR, Bandaru P, Templeton S, Lee D, Karandur D, Pelton JG, Marqusee S, Wemmer D, Kuriyan J | 2022 | A saturation-mutagenesis analysis of the interplay between stability and activation in Ras | https://doi.org/10.5281/zenodo.6131510 | Zenodo, 10.5281/zenodo.6131510 |

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
