## [Editor Report]

This is a well executed study that provides significant new insights and connections between protein structure, stability and function. Numerous sites of mutation are identified that activate the small GTPase Ras beyond the few cancer "hot spots" that predominate cancer genomics data. While cancer causing mutations selectively alter regulatory interactions with GTPase-activating proteins, careful biophysical analysis presented here leads to the conclusion that Ras is also activated by mutations that decrease stability (increase dynamics) short of unfolding. Thus, protein sensitivity to activating mutations can depend on the stability threshold in addition to regulatory interactions with binding partners.

---

## [Decision Letter]

**Decision letter after peer review:**

Thank you for resubmitting the paper entitled, "A Saturation-Mutagenesis Analysis of the Interplay Between Stability and Activation in Ras", has been reviewed by three peer reviewers, including Amy Andreotti as the Reviewing Editor and Reviewer #, and the evaluation has been overseen by a Senior Editor. The following individual involved in review of your submission has agreed to reveal their identity: Nicola Salvi (Reviewer #3).

*Reviewer #1:*

This manuscript builds on prior work from this group that found multiple activating Ras mutations in a high-throughput single-site saturation mutagenesis bacterial screen. The result was unexpected given that Ras mutations in the cancer databases are largely limited to three positions: G12, G13 and Q61. The results of detailed saturation mutagenesis experiments carried out in mammalian and bacterial cells, with and without regulators, and focusing on full-length versus a truncated Ras construct convincingly demonstrate that the earlier results in bacteria were not an artifact. While the findings are specific to Ras signaling, the broader finding that the simpler bacterial assay is reliable should be important for a range of systems.

Having established that the activating mutations arising from saturation mutagenesis experiments are robust, the authors next examine the stability of wild type and mutant Ras. Interestingly, two of the three cancer ‘hot-spot’ mutations show no difference in stability compared to wild type and the third is only slightly destabilized in these assays. In contrast, many of the other mutations that arose in the screens show significant destabilization and the authors link these observations with increased activity and nicely describe the idea of “stability-dependent gain-of-function” mutants. The authors next demonstrate the predictive strength of the saturation-mutagenesis approach to identify mutations that appear in cancer databases and draw the conclusion that evading GAP inactivation is a crucial feature of Ras cancer mutations.

Overall, this study is a rigorous and thorough analysis of how saturation mutagenesis can be used to probe a given sequence for activating sites. The findings lead to in-depth consideration of the interplay between stability and activity.

*Reviewer #2:*

The authors continued their bacterial two-hybrid assay to study the mutational profile of H-Ras in mammalian Ba/F3 cell. The results indicated that the mutational landscape of mammalian-expressed Ras resembles those of the bacterial assay when a GAP is co-expressed.

This result, the outcome of a great effort and much time, should be of interest to the community. The work is innovative and impressive, and lends itself to thoughtful extensions and applications to further its impact in a vastly important area.

*Reviewer #3:*

The aim of the authors is to understand the mutational spectrum of Ras in cancer. To do so, they use single-site saturation mutagenesis in both bacterial and mammalian cells. While both sets of data reveal that gain-of-function mutations include – but are absolutely not limited to – mutations common in cancer, a careful analysis of the differences between the two screens allows the authors to formulate hypotheses about the mechanism underlying the activity of distinct classes of mutations. Very conveniently, these hypotheses are tested characterising protein stability using a range of complementary techniques, including NMR and MS.

The results indicate that while most non-oncogenic gain-of-function mutations result in higher flexibility and consequently faster nucleotide exchange, cancer hotspot mutations are unique in that they prevent successful Ras activation by GTPase-activating proteins (GAPs).

These findings should stimulate the field to better characterise at the molecular and structural level the GAP:Ras interaction. It remains to be explored whether further studies of this interaction will enable the use of the GAP:Ras interaction as a target in cancer treatment.

The work is well written, and the data support the conclusions almost completely, although some additional data could strengthen the conclusion even further (see point 3 below). The following are suggestions for the authors:

1) Not enough information is provided to estimate the impact of the point mutations on the HSQC spectra of the variants. One representative assigned spectrum for each variant + the WT should be included, to make it possible to assess how robustly the authors could port the WT assignment to the mutants. It would also be good if the authors could indicate how many residues could actually be assigned for each mutant.

2) The R2 > 0.7 threshold for including decays in the NMR analysis appears rather arbitrary. Why this particular choice? Wouldn't it be possible to use all "EX2" residues from the MS analysis? At the very least, the authors should use a statistically sound model testing protocol to justify their choice of excluding some data from the analysis.

3) I regret the lack of any NMR and/or MS data on the HDX properties of oncogenic variants. Similarly, such data are also not present for long vs short WT Ras. These additional data points would be interesting because absence of changes in thermal stability does not exclude per se enhanced flexibility (see for example work of Vihinen in the 80s). I think that the paper would have a significantly higher impact if the authors could include at least some of these additional data.

4) The authors may consider extending their discussion. Other than restating the main point of the link between the stability of Ras and the fitness profile of single residue mutations, there are several potential considerations/applications of the scheme developed in their studies. The first relates to computational assessment of the predicted mutations. To assess how a single mutation affects the population shift of an oncogenic protein conformation, the overall stability of the protein after the mutation needs to be considered. This would yield the protein functional expression level. Second, the experimental scheme can be used to reveal how the oncogenic mechanism of the mutations, e.g., the predominant Ras hotspots at Gly 12, Gly 13, and Gln 61 in cancer databases prevent Ras inactivating by GAP hydrolysis of the bound GTP to GDP. Third, although frequent cancer mutations likely correspond to driver mutations, infrequent variant mutations may not always correspond to passengers. Thus, the experimental scheme can also verify infrequent mutations as a category of driver or "latent" driver mutation, which should interest the pharmaceutical industry. Lastly, if the authors can summarize how to generalize their remarkable approach to apply to all proteins that contribute to cancer initiation, progress, and metastasis, it might further highlight the contribution of the paper to the field of cancer mutation studies.

5) Given the emphasis on protein destabilization/unfolding, more discussion about the possible role of ubiquitination could strengthen the discussion. Is it possible that one reason the destabilizing mutations discovered here are not prominent in cancer databases may be that the mutation-induced destabilization makes these variants more prone to proteasomal degradation in the cell?

---

## [Author Response]

Reviewer #3:The aim of the authors is to understand the mutational spectrum of Ras in cancer. To do so, they use single-site saturation mutagenesis in both bacterial and mammalian cells. While both sets of data reveal that gain-of-function mutations include – but are absolutely not limited to – mutations common in cancer, a careful analysis of the differences between the two screens allows the authors to formulate hypotheses about the mechanism underlying the activity of distinct classes of mutations. Very conveniently, these hypotheses are tested characterising protein stability using a range of complementary techniques, including NMR and MS.The results indicate that while most non-oncogenic gain-of-function mutations result in higher flexibility and consequently faster nucleotide exchange, cancer hotspot mutations are unique in that they prevent successful Ras activation by GTPase-activating proteins (GAPs).These findings should stimulate the field to better characterise at the molecular and structural level the GAP:Ras interaction. It remains to be explored whether further studies of this interaction will enable the use of the GAP:Ras interaction as a target in cancer treatment.The work is well written, and the data support the conclusions almost completely, although some additional data could strengthen the conclusion even further (see point 3 below). The following are suggestions for the authors:1) Not enough information is provided to estimate the impact of the point mutations on the HSQC spectra of the variants. One representative assigned spectrum for each variant + the WT should be included, to make it possible to assess how robustly the authors could port the WT assignment to the mutants. It would also be good if the authors could indicate how many residues could actually be assigned for each mutant.

As noted in the manuscript, the wild-type spectrum was assigned by comparison to a published reference assignment. For each mutant, the HSQC was calibrated based on assignments that were far from the mutation site based on both structure and sequence. After this, the peak assignments were assessed one by one, and any non-overlapping peaks were ignored. Since we did not rely on complete assignments, this conservative approach was deemed to be reliable. For each construct, the number of peaks assigned is listed below. Due to the high density of information in the manuscript, we have decided not to include additional spectra and data in the manuscript.

**Author response table 1. sa2table1:** 

Construct	Wild-type	L120A	Y157Q	H27G	Q99A
Number of peaks	127	109	104	82 (after 1 hour of deuterium exchange)	113

2) The R2 > 0.7 threshold for including decays in the NMR analysis appears rather arbitrary. Why this particular choice? Wouldn't it be possible to use all "EX2" residues from the MS analysis? At the very least, the authors should use a statistically sound model testing protocol to justify their choice of excluding some data from the analysis.

The MS does not have residue by residue resolution so it is not possible to select EX2 residues on a residue-by-residue basis. We instead eliminated any residues from the NMR analysis that were overlapping with peptides that demonstrated EX1 behavior. Other residues that were eliminated were likely exchanging either too fast or too slow to be analyzed by the NMR experiments. In the case that they were exchanging too slow, these peaks still remain unchanged after 24 hours in D_2_O and therefore, their rate of change cannot be fit to a single exponential. These peaks result in a low R^2^ when fit to a single exponential. Residues that exchange too fast, have disappeared by the time the first HSQC has been recorded, and thus cannot be tracked in the subsequent HSQC spectra. Some peaks did not disappear until a later time point, but were still difficult to fit due to the lack of points to fit to a decay, and therefore also resulted in a poor R^2^. Based on this observation, a cutoff needed to be established that would eliminate these peaks from the analysis. A histogram of the R^2^ values for representative experiments for each mutant shows a minimum around 0.5, which perhaps would be a reasonable cutoff. However, after looking at the exchange data for peaks that would fall into this category, we decided that was not stringent enough. An R^2^ cutoff of 0.7 is more stringent, and allows for a good fit even if one or two points at the baseline (when the peak has almost disappeared) are noisy.

**Author response image 1. sa2fig1:** 

3) I regret the lack of any NMR and/or MS data on the HDX properties of oncogenic variants. Similarly, such data are also not present for long vs short WT Ras. These additional data points would be interesting because absence of changes in thermal stability does not exclude per se enhanced flexibility (see for example work of Vihinen in the 80s). I think that the paper would have a significantly higher impact if the authors could include at least some of these additional data.

We agree with the reviewer that such data would indeed be interesting to obtain, and such experiments are planned but cannot be completed in a suitable timeframe for inclusion in this publication.

4) The authors may consider extending their discussion. Other than restating the main point of the link between the stability of Ras and the fitness profile of single residue mutations, there are several potential considerations/applications of the scheme developed in their studies. The first relates to computational assessment of the predicted mutations. To assess how a single mutation affects the population shift of an oncogenic protein conformation, the overall stability of the protein after the mutation needs to be considered. This would yield the protein functional expression level. Second, the experimental scheme can be used to reveal how the oncogenic mechanism of the mutations, e.g., the predominant Ras hotspots at Gly 12, Gly 13, and Gln 61 in cancer databases prevent Ras inactivating by GAP hydrolysis of the bound GTP to GDP. Third, although frequent cancer mutations likely correspond to driver mutations, infrequent variant mutations may not always correspond to passengers. Thus, the experimental scheme can also verify infrequent mutations as a category of driver or "latent" driver mutation, which should interest the pharmaceutical industry. Lastly, if the authors can summarize how to generalize their remarkable approach to apply to all proteins that contribute to cancer initiation, progress, and metastasis, it might further highlight the contribution of the paper to the field of cancer mutation studies.

We appreciate the support for our work that is expressed in these comments. Rather than alter the current manuscript, we have decided to reserve these ideas for inclusion in a subsequent manuscript.

5) Given the emphasis on protein destabilization/unfolding, more discussion about the possible role of ubiquitination could strengthen the discussion. Is it possible that one reason the destabilizing mutations discovered here are not prominent in cancer databases may be that the mutation-induced destabilization makes these variants more prone to proteasomal degradation in the cell?

We agree with this comment, and had already mentioned this as a possibility in the closing statements of the manuscript. It is certainly an avenue that we will explore more fully in the future.